# Limited role of generation time changes in driving the evolution of the mutation spectrum in humans

Ziyue Gao[1]*, Yulin Zhang[2], Nathan Cramer[3], Molly Przeworski[4,5], Priya Moorjani[2,3]*

[1]Department of Genetics, University of Pennsylvania, Perelman School of Medicine, Philadelphia, United States; [2]Center for Computational Biology, University of California, Berkeley, Berkeley, United States; [3]Department of Molecular and Cell Biology, University of California, Berkeley, Berkeley, United States; [4]Department of Biological Sciences, Columbia University, New York, United States; [5]Department of Systems Biology, Columbia University, New York, United States

**Abstract** Recent studies have suggested that the human germline mutation rate and spectrum evolve rapidly. Variation in generation time has been linked to these changes, though its contribution remains unclear. We develop a framework to characterize temporal changes in polymorphisms within and between populations, while controlling for the effects of natural selection and biased gene conversion. Application to the 1000 Genomes Project dataset reveals multiple independent changes that arose after the split of continental groups, including a previously reported, transient elevation in TCC>TTC mutations in Europeans and novel signals of divergence in C>Gand T>A mutation rates among population samples. We also find a significant difference between groups sampled in and outside of Africa in old T>C polymorphisms that predate the out-of-Africa migration. This surprising signal is driven by TpG>CpG mutations and stems in part from mis-polarized CpG transitions, which are more likely to undergo recurrent mutations. Finally, by relating the mutation spectrum of polymorphisms to parental age effects on de novo mutations, we show that plausible changes in the generation time cannot explain the patterns observed for different mutation types jointly. Thus, other factors – genetic modifiers or environmental exposures – must have had a non-negligible impact on the human mutation landscape.

*For correspondence:
ziyuegao@pennmedicine.upenn.edu (ZG);
moorjani@berkeley.edu (PM)

## Editor's evaluation

This important study investigates temporal variation in patterns of germline mutation during the evolution of human populations. Using a compelling approach that controls for the effects of selection and biased gene conversion the authors show that changes in generation time alone cannot explain the joint patterns observed for different mutation types, suggesting that other factors such as genetic modifiers or environmental exposures must have played a role as well. This work will be of broad interest to population geneticists and evolutionary biologists.

## Introduction

Recent advances in high-throughput sequencing have enabled large-scale surveys of genetic variation in thousands of humans, providing a rich resource for understanding the source and mechanisms shaping the mutation landscape over time. Comparisons of polymorphism patterns across geographic population samples have uncovered numerous differences in the mutation rates and spectra (i.e., relative proportions of different types of mutations) (*DeWitt et al., 2021*; *Goldberg and Harris, 2022*;

**eLife digest** Each human has 23 pairs of chromosomes, one set inherited from each parent. But the child's chromosomes are not an exact copy of their parents' chromosomes. Spontaneous changes or mutations in the DNA during the formation of the egg or sperm cells, or early development of the embryo, can change a small fraction of the nucleotides or 'letters' that make up the DNA. These modifications are an important source of genetic diversity in human populations and contribute to the evolution of new traits.

Each genetic variant in present-day human populations represents a mutation in one of their ancestors. The types and frequencies of variants vary across human populations and have changed over time, suggesting that mutation patterns have evolved in the past. But the processes driving these population-level differences remain elusive. One possible factor may be changes in the average age of reproduction or the generation time in a population . For example, older parents contribute more – and also different types of – new mutations to their children than younger parents do. Populations, where it is customary to have children at older ages, may therefore have a different mutation landscape.

To find out if this is indeed the case, Gao et al. used computer algorithms to analyze the genomes of hundreds of people living on three continents who participated in 'the 1,000 Genomes Project'. The analysis identified differences in mutation patterns across continental groups and estimated when these changes occurred. Further, they showed that although the age of reproduction had an impact on the mutation landscape, differences in generation time alone could not explain the observed changes in the human mutation spectrum. Factors other than generation time, such as environmental exposures, may have played a role in shifting these patterns.

The study provides new insights into the changes in the mutation landscape over the course of human evolution. Mapping these patterns in humans worldwide may help scientists understand the causes underlying these changes. The techniques used by Gao et al. may also help analyze changes in mutation patterns in other organisms.

*Harris, 2015*; *Harris and Pritchard, 2017*; *Hwang and Green, 2004*; *Mathieson and Reich, 2017*; *Moorjani et al., 2016a*; *Narasimhan et al., 2017*; *Speidel et al., 2019*). A notable signal in humans is the enrichment of TCC>TTC variants in polymorphism data from Europeans relative to Africans and Asians (*Harris and Pritchard, 2017*). This signal is also observed in South Asians to a lesser degree and has been suggested to originate in ancient Neolithic farmers (*Harris and Pritchard, 2017*; *Speidel et al., 2021*). Many other subtle but statistically significant signals have also been detected; given the recent common ancestry of human populations, this finding indicates that the mutational spectrum in humans has been evolving rapidly.

Several genetic and nongenetic factors have been implicated as affecting mutation rates and acting as potential drivers of observed interpopulation differences in the mutation spectrum of polymorphisms. First, some environmental exposures can increase mutation rates, especially of particular types. As humans in different geographic locations and environments may have experienced differential exposures over the past 50,000–100,000 years since the out-of-Africa (OOA) migration, rates of specific mutation types could have diverged between populations (*Harris, 2015*; *Mathieson and Reich, 2017*). Second, genetic modifiers of mutation rates, such as variants in genes that copy or repair DNA, could segregate at different frequencies across populations. Despite the deleterious effects of alleles that modify mutation rates, in recombining species, they could be nearly neutral and maintained for a long time, leading to genome-wide differences across populations (*Milligan et al., 2022*; *Seoighe and Scally, 2017*).

In addition, direct sequencing of human pedigrees has revealed the effects of the parental ages at reproduction on the relative fractions of mutation types (*Goldmann et al., 2018*; *Jónsson et al., 2017*). For example, as parents age, fathers pass on disproportionally more T>C mutations, and mothers contribute a higher fraction of C>G mutations (*Jónsson et al., 2017*). Thus, differences in the average reproductive ages, or equivalently 'generation times,' alone can lead to differences in mutation spectrum across populations; indeed, such differences have been invoked to explain a large fraction of observed variation in types of polymorphisms among population samples (*Coll Macià et al., 2021*).

The joint distribution of mutation type and frequency of polymorphisms, however, depends not only on the mutational input, but also on other evolutionary forces such as natural selection, biased gene conversion, and demography. In particular, natural selection distorts the allele frequency distribution and fixation probability of non-neutral variants, and the average effect of natural selection can differ across mutation types (*Wakeley, 2010*). As an example, genic regions tend to be more GC-rich, so mutations at G:C base pairs may be subject to stronger purifying or background selection compared to mutations at A:T base pairs (*Lander et al., 2001*; *McVicker et al., 2009*). GC-biased gene conversion (gBGC) is another process that exerts differential effects across mutation types by effectively acting like positive selection favoring mutations from weak alleles (A or T) to strong alleles (C or G) and negative selection against mutations from strong to weak alleles (*Duret and Galtier, 2009*). The strengths of selection and gBGC depend on the effective population size and thus on the demographic history of a population. Demographic history also influences allele frequencies for a given allele age (*Kimura, 1969*). This poses a challenge in interpreting previous studies (*Harris and Pritchard, 2017*; *Mathieson and Reich, 2017*) aimed at learning about when changes in mutational processes may have occurred by using allele frequencies, as mutations of the same frequency can have drastically different distributions of ages in distinct populations (e.g., doubletons in Africans are substantially older than doubletons in Europeans or Asians; *Mathieson and McVean, 2014*).

Beyond the biological processes that shape polymorphism data, the characterization of the mutational spectrum can be biased by many technical issues. For instance, a recent study showed that some interpopulation differences discovered in low-coverage 1000 Genomes data may be driven by cell line artifacts or errors in PCR amplification (*Anderson-Trocmé et al., 2020*). Further, comparisons of mutation patterns across datasets are sensitive to differences in the accessible genomic regions across studies. Because there is large variation in mutation rates and base pair composition across genomic regions, differences in the regions sequenced across studies can have a non-negligible impact on comparisons of mutation spectrum across datasets (*Monroe et al., 2022*; *Seplyarskiy et al., 2021*). In addition, the number of genomes surveyed, in combination with the specific population demographic history, influences the chance of observing repeated mutations at the same site, and thus the observed polymorphism patterns (*Lek et al., 2016*). Given these challenges, it remains unclear whether the numerous observed differences across human populations stem from rapid evolution of the mutation process itself, other evolutionary processes, or technical factors.

Motivated by these considerations, we propose a new framework to compare the mutation spectrum over time and across human populations. First, we infer the age of each derived allele observed in a population using a newly developed approach, Relate, which reconstructs local genealogies and estimates allele ages (*Speidel et al., 2019*). This approach allows us to perform more reliable comparisons across populations as well as to investigate changes in mutation processes across time. Next, we minimize confounding effects of selection by removing constrained regions and known targets of selection in the genome. We also control the effects of biased gene conversion by focusing on comparison of pairs of mutations (e.g., T>C and T>G) that are subject to similar effects of gBGC. This pairwise comparison further mitigates the issue of interdependencies in comparing mutation fractions (i.e., an increased contribution of one mutation type necessarily lowers the contribution of other mutation types). Based on this new framework, we re-evaluate the evidence for evolution of the mutation spectrum in human populations and investigate when, how, and in which populations significant changes have occurred over the course of human evolution. Finally, by relating parental age effects on the mutation spectrum estimated in contemporary pedigrees to the observed patterns of polymorphisms of varying ages, we evaluate the role of changes in generation times in shaping the human mutation landscape.

## Results

### Variation in the spectrum of human polymorphisms over time

We analyzed single-nucleotide polymorphisms (SNP) identified in high-coverage whole-genome sequencing data from the 1000 Genomes Project, including 178 individuals of West African ancestry living in Ibadan, Nigeria (YRI), 179 individuals of Northern European ancestry living in the United States (CEU), and 103 individuals of East Asian ancestry living in Beijing, China (CHB) (*Byrska-Bishop et al., 2022*). To focus on putatively neutral mutations, we removed exons and phylogenetically conserved

regions as previous studies (*Harris and Pritchard, 2017*; *Moorjani et al., 2016a*). To perform reliable comparison between datasets in downstream analysis and ensure the results are not driven by local genomic differences in mutation rate, we focused on regions that were accessible in both population and pedigree datasets ( hereafter, referred to as 'commonly accessible regions') ('Materials and Methods').

We inferred the age of each derived variant (with the ancestral allele determined based on the six primate EPO (Enredo, Pecan, Ortheus) alignment) in YRI, CEU, and CHB using Relate, a method to reconstruct local genealogies based on phased haplotype sequences (*Speidel et al., 2019*). We then divided all SNPs into 15 bins based on the ages of the derived allele inferred by Relate, accounting for uncertainty in the inferred mutation age by assuming a uniform distribution of ages between the inferred lower and upper bounds for each variant ('Materials and methods'). We classified each SNP into six disjoint classes based on the type of base pair substitution: T>A, T>C, T>G, C>A, C>G, and C>T (each including the corresponding substitution on the reverse complement strand, e.g., T>C includes both T>C and A>G substitutions). Given the well-characterized hypermutability of methylated CpG sites (*Duncan and Miller, 1980*; *Kong et al., 2012*), we further divided C>T SNPs into subtypes occurring in CpG and non-CpG contexts by considering the flanking base pair on either side of the variant.

We find marked differences in the relative proportions of different mutation types (i.e., the mutation spectrum) across varying allele age bins within CEU (*Figure 1*) as well as in YRI and in CHB (*Figure 1—figure supplement 1*), as seen earlier in the low-coverage 1000G data (*Speidel et al., 2019*). We obtain qualitatively similar results when considering other 1000G populations of TSI, LWK, and JPT (*Figure 1—figure supplement 1*). This observation echoes previous findings about the evolution of the mutation spectrum comparing polymorphisms across allele frequencies (*Carlson et al., 2018*; *Harris and Pritchard, 2017*; *Mathieson and Reich, 2017*). As noted previously, however, differences in mutation spectrum across frequencies alone are weak evidence for the evolution of the mutation process itself because patterns of standing polymorphisms can be affected by repeat mutations and other evolutionary forces, including selection and gene conversion.

Notably, the infinite sites model is a reasonable assumption for small sample sizes (*Kimura, 1969*), but recurrent mutations become highly likely in large datasets, especially at sites with higher mutation rates (*Harpak et al., 2016*; *Lek et al., 2016*). Recurrent, multi-allelic, and back mutations violate the model assumptions of Relate and are often excluded from its output. For instance, given the higher mutation rate of transitions at CpG sites, such SNPs are more likely to be subject to recurrent mutations in a large sample and thus may map to multiple branches in the tree, leading to their exclusion from Relate's output (*Speidel et al., 2019*). As expected from these considerations, the fraction of CpG C>T SNPs in young mutations (i.e., those estimated to have occurred in the past ~50 generations) is lower than proportions in de novo mutations (DNMs) in present-day pedigree studies (*Figure 1—figure supplement 2*). Differences in mutation spectrum across age bins in modern humans persist even after excluding CpG C>T mutations (*Figure 1—figure supplement 3*), however, indicating that other mutation types are also changing in relative frequency over time and the observed patterns are not driven solely by recurrent mutation at CpG sites.

Next, we examined the effect of linked selection on different mutation types. While we excluded direct targets of selection from analysis (i.e., exons and conserved regions), much of the genome is linked to non-neutral variants and subject, notably, to background selection (*Charlesworth et al., 1993*; *McVicker et al., 2009*; *Murphy et al., 2022*). A common measure of the effects of background selection is the *B*-statistic or *B*-score that estimates the reduction in nucleotide diversity levels compared to the neutral expectation (*McVicker et al., 2009*). To characterize the impact of linked selection, we calculated the average genome-wide *B*-score of each mutation type. We find nearly identical average *B*-scores and similar distributions for all mutation types (*Figure 1—figure supplement 4*). Further, comparing the mutation spectrum over time in CEU, YRI, and CHB, we obtain qualitatively similar results when restricting to regions with weak background selection (*B*-score > 800, where the genetic diversity is reduced by <20% compared to the neutral expectation; *Figure 1—figure supplement 5*). These analyses suggest that although linked selection has pervasive effects, its average impact is relatively uniform across the seven mutation types in commonly accessible regions ('Materials and methods).

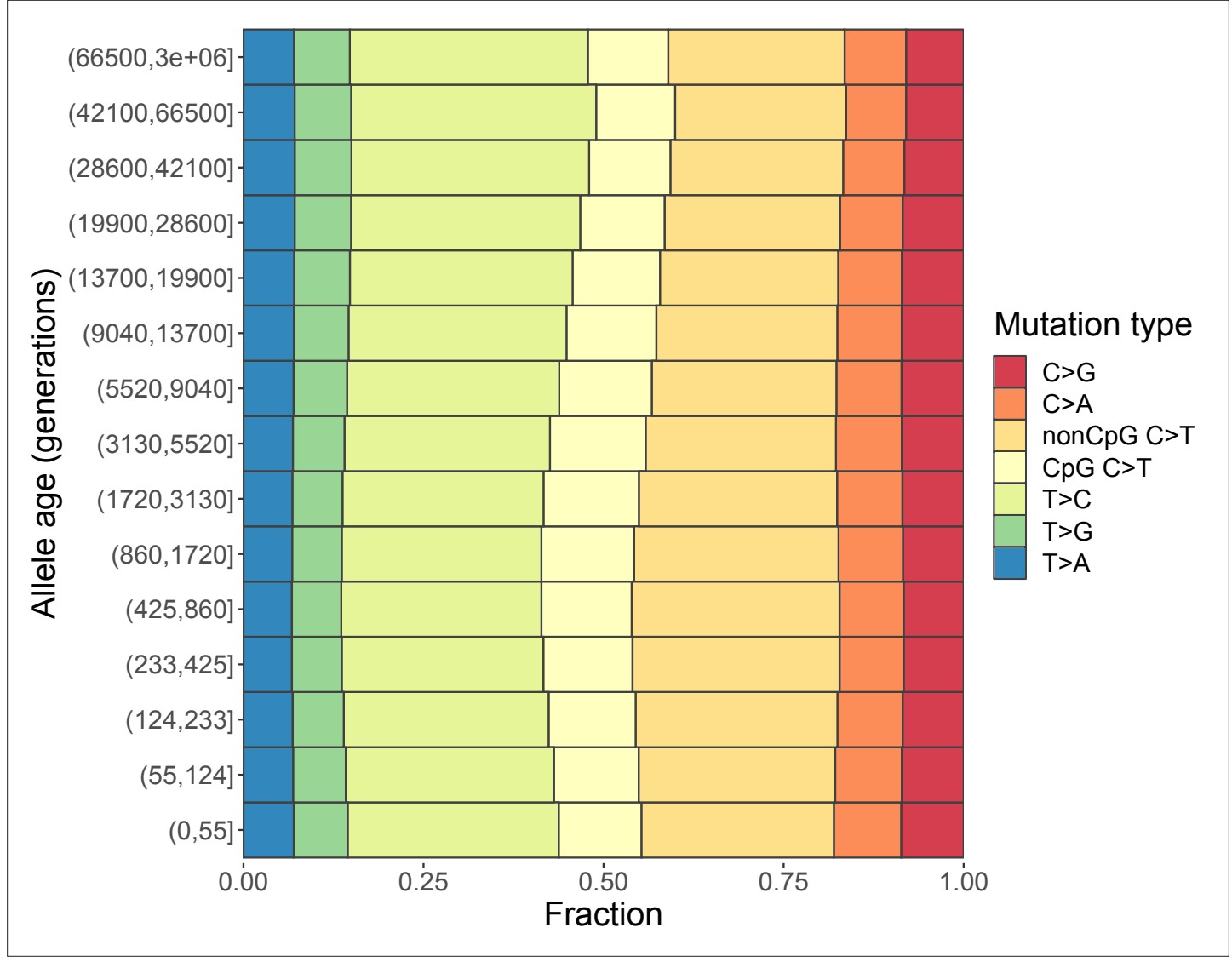

**Figure 1.** Changes in the mutation spectrum of polymorphisms in CEU over evolutionary time.

The online version of this article includes the following source data and figure supplement(s) for figure 1:

**Source data 1.** Bedfile for the commonly accessible region excluding exons and phylogenetically conserved elements.

**Source data 2.** Text files with (pseudo-)counts of different types of mutations in YRI, LWK, CEU, TSI, CHB, JPT in each time window.

**Figure supplement 1.** Mutation spectra of polymorphisms stratified by allele age in six human populations (YRI, LWK, CEU, TSI, CHB, and JPT).

**Figure supplement 2.** Mutation spectra of CEU polymorphisms and de novo mutations (DNMs) from Icelandic trios (*Halldorsson et al., 2019*).

**Figure supplement 3.** Mutation spectra of CEU polymorphisms and de novo mutations (DNMs) from Icelandic trios, excluding C>T transitions at CpG sites.

**Figure supplement 4.** Mean and distribution of B-scores of different mutation types.

**Figure supplement 5.** Mutation spectra of human polymorphisms in genomic regions with weak background selection (*B*-score > 800).

**Figure supplement 6.** Fractions of S>S, S>W, W>S, and W>W mutations stratified by (**A**) derived allele frequency and (**B**) allele age.

**Figure supplement 7.** Distribution of variants subject to biased gene conversion stratified by allele age and recombination rate.

**Figure supplement 8.** Mutation spectra of human polymorphisms in six populations (YRI, CEU, CHB, LWK, TSI, and JPT) stratified by allele age based on alternative binning strategies.

Gene conversion is another evolutionary process that can have a profound impact on the mutation spectrum of polymorphisms. gBGC acts like selection for certain mutation types by causing the preferential transmission of strong (S) alleles (C or G) over weak (W) alleles (A or T) in heterozygotes (*Duret and Galtier, 2009*). Accordingly, we observe enrichments of W>S mutations (T>C and T>G) in common variants and of S>W mutations (C>A and C>T) in rare variants (*Figure 1—figure supplement 6A*). Moreover, gBGC violates model assumptions of Relate (for both neutrality and infinite-sites mutation model) and could lead to subtle biases in estimated allele ages (*Speidel et al., 2019*). Due to the effect of gBGC, W>S mutations are expected to be enriched in older variants compared to S>W variants, and this enrichment is expected to be stronger in regions with high recombination rates (*Glémin et al., 2015*). Indeed, we observe such enrichment and the expected correlation with recombination rate (*Figure 1—figure supplement 7A*), supporting the effect of gBGC on the mutation spectrum of variants across mutation ages. Furthermore, the effect of gBGC is expected to vary across populations as its strength depends on the effective population size. Accordingly, we observe that the trends of the ratio of W>S to S>W over time differ across human populations (*Figure 1—figure supplement 6B*, *Figure 1—figure supplement 7B*). These results highlight the need to account for gBGC in order to reliably interpret the source of observed differences within and between populations (whether using allele frequency bins or allele age estimates).

## Pairwise comparisons of mutation types accounting for gBGC

In light of the impact of gBGC on the mutation spectrum, we focused on comparisons of pairs of mutation types subject to similar effects of gBGC (i.e., in which both are favored, disfavored, or unaffected by gBGC). Specifically, we focused on four pairwise comparisons including (1) C>T at non-CpGs vs. C>A at non-CpGs; (2) C>T CpGs vs. C>A CpGs; (3) C>G vs. T>A; and (4) T>C vs. T>G. In principle, it is possible that the strength of gBGC is distinct for different types of variantsinvolving S and W alleles (*Tsai-Wu et al., 1992*). However, in mice, roughly similar conversion rates are observed for C>A and C>T non-crossover gene conversion events as well as for T>C and T>G events (*Li et al., 2019*), lending support to using pairwise mutation ratios for controlling the effects of gBGC at least to a first approximation.

Three of the four pairwise comparisons involve mutation types with the same mutational opportunity (e.g., both T>C and T>G mutations involve changes at ancestral T bases in the genome), which further minimizes the confounding effects of regional variation on the chance of recurrent mutation or strength of background selection. Moreover, the pairwise ratios impose no co-dependency among mutation types as the four comparisons are mathematically independent of each other (although they may be biologically dependent if multiple ratios are affected simultaneously by some change in the mutational process).

Investigating the mutation spectrum using these four pairwise comparisons, we observe marked differences in the ratios both over evolutionary time and across populations. Specifically, we find multiple independent signals of mutation rate evolution, reflected by both temporal variation within a population and differences between YRI, CHB, and CEU ($p<0.01$ by chi-square test after correcting for multiple hypothesis testing; 'Materials and methods'; *Figure 2A*). These differences may represent broader geographic or population differences as we replicate these findings in other population samples from the same continents – LWK, TSI, and JPT – from the 1000 Genomes Project (*Figure 2—figure supplement 1*).

We performed multiple sanity checks to rule out any technical artifacts or sources unrelated to the mutation process in contributing to the observed interpopulation differences. We obtain qualitatively similar results when restricting the analysis to putatively neutral regions with *B*-score > 800 (*Figure 2—figure supplement 2*) or comparing regions with high and low recombination rates (*Figure 2—figure supplement 3*), confirming that the use of pairwise comparisons effectively controls for the effects of selection and gBGC. In turn, to account for potential inaccuracies in mutation ages estimated by Relate, we stratified variants by allele frequencies instead of inferred mutation ages and replicated the signals detected in mutation age analysis (*Figure 2—figure supplement 4*). We again observe similar results with different binning strategies for allele age, based on inferred mutation ages in YRI or CHB (*Figure 2—figure supplement 5*). Together, these results provide strong evidence that the human germline mutation spectrum has evolved over time and differs across populations.

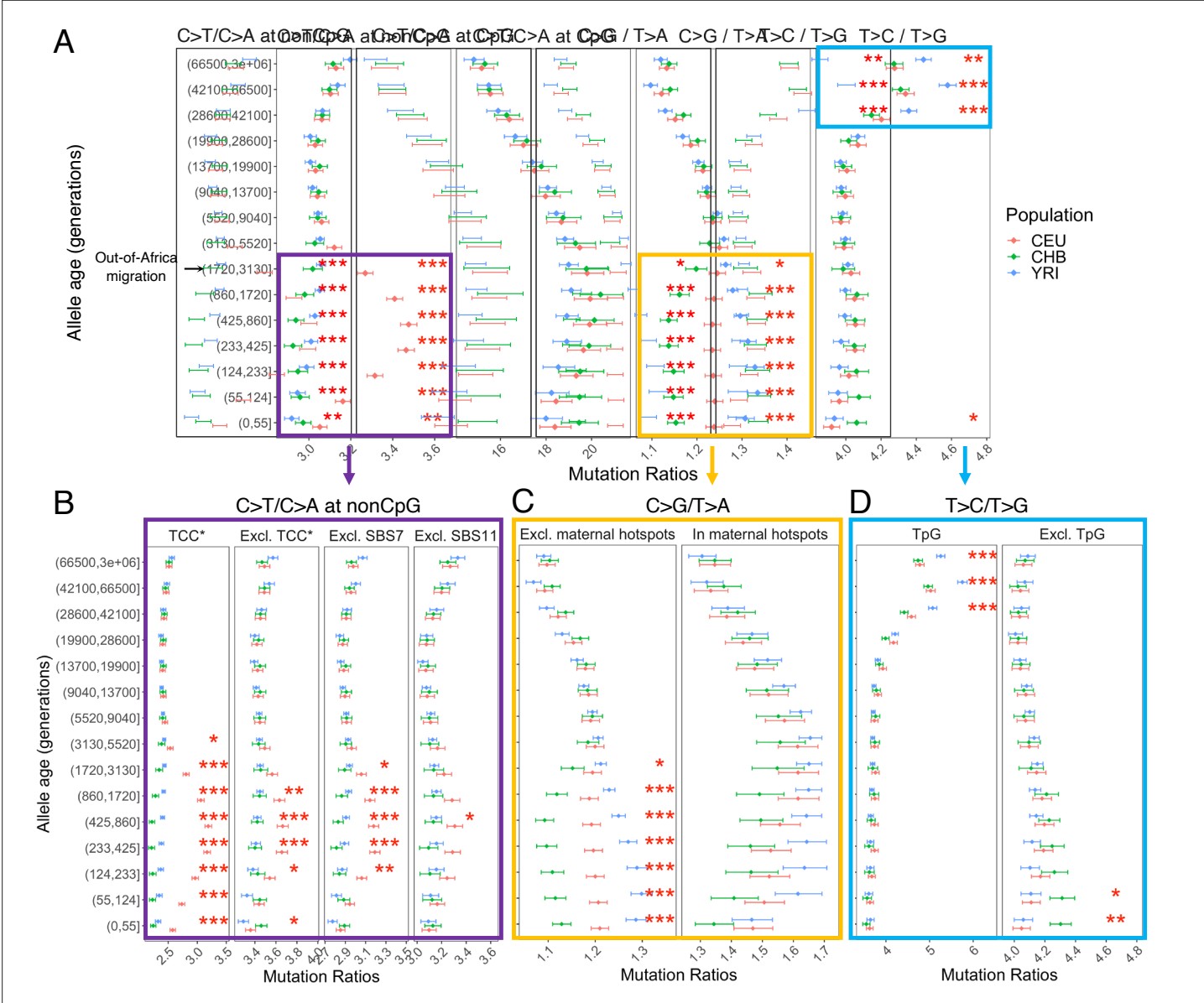

**Figure 2.** Comparison of pairwise mutation ratios for polymorphisms arising in different time windows. (**A**) Four pairwise mutation ratios are shown, each of which compares two mutation types that are matched for mutational opportunity and effects of GC-biased gene conversion (gBGC). The black arrow indicates the window coinciding with the out-of-Africa (OOA) migration. The points represent the observed polymorphism ratios, while the whiskers denote the 95% CI assuming a binomial distribution of polymorphism counts. Highlighted in boxes are three ratios that show significant interpopulation differences, with in-depth investigation into each shown in lower panels. Asterisks refer to the p-value obtained from a chi-square test after a Bonferroni correction for 60 tests: *p<0.01, ** p<0.0001 and ***p<10[-8] (same indicators of significance levels were used in Figure supplements). (**B**) Elevation in C>T/C>A ratio in CEU at non-CpG sites, after excluding the four trinucleotide contexts (TCC, TCT, CCC, and ACC) previously identified to be associated with the TCC pulse in Europeans (denoted by TCC*; *Harris and Pritchard, 2017*), as well as contexts affected by Catalog of Somatic Mutations in Cancer (COSMIC) mutational signatures of SBS7 and SBS11 (*Harris, 2015*; *Mathieson and Reich, 2017*). (**C**) Post-OOA divergence in C>G/T>A ratio among three population groups. (**D**) Higher T>C/T>G ratio in YRI than CEU and CHB samples among extremely old variants, driven by TpG variants.

The online version of this article includes the following source data and figure supplement(s) for figure 2:

**Source data 1.** Text files with (pseudo-)counts of mutations classified into eight types in genomic regions including, excluding, and within the maternal C>G mutation hotspots, in YRI, LWK, CEU, TSI, CHB, and JPT in each time window.

**Figure supplement 1.** Pairwise polymorphism ratios in YRI, CEU and CHB, as well as three additional populations of African (LWK), European (TSI), and East Asian (JPT) ancestry in the 1000 Genomes Project.

**Figure supplement 2.** Pairwise polymorphism ratios in genomic regions with weak background selection (*B*-score >800).

*Figure 2 continued on next page*

Below we discuss the timing, direction, and population origin of the mutation rate changes related to each of the signals we detected in detail.

## Elevation of non-CpG C>T/C>A ratio in Europeans

The largest signal that we observed is the transient elevation in the ratio of C>T/C>A mutations at non-CpG sites in CEU compared to the ancestral state before the OOA migration; in contrast, the non-CpG C>T/C>A ratios of CHB and YRI do not exhibit a similar shift at recent timescales. This signal encompasses the previously reported enrichment of C>T polymorphisms in a TCC context in Europeans, as well as other trinucleotide contexts (*Harris and Pritchard, 2017*; *Mathieson and Reich, 2017*; *Speidel et al., 2019*). Investigating the temporal patterns in CEU, we find that the increase in the ratio of C>T/C>A mutations at non-CpG sites becomes discernible starting from the time window spanning the OOA migration (50,000–100,000 years ago or ~2000–4000 generations ago) (*Schiffels and Durbin, 2014*), peaks around 238–887 generations ago, and subsides in the most recent age bin of 0–55 generations (*Figure 2A*). Because there is large uncertainty in inferred allele ages and our binning approach often spreads the contribution of each variant over two or more age bins ('Materials and methods'), the timeline and magnitude of variation should be interpreted cautiously: the transient change in non-CpG C>T mutations was likely shorter-lived and possibly of higher intensity than our results suggest. However, the temporal and geographic enrichment patterns from our analysis are consistent with previous reports based on low-coverage 1KG or other datasets (*Harris and Pritchard, 2017*; *Mathieson and Reich, 2017*; *Speidel et al., 2019*).

Among all non-CpG trinucleotide contexts, the interpopulation differences are most pronounced in the four previously reported trinucleotide contexts (TCC, TCT, CCC, and ACC; *Harris and Pritchard, 2017*), but are detectable in other non-CpG contexts as well (*Figure 2B*). Previous analysis found that these mutational contexts are enriched in two of the mutational signatures extracted from somatic mutations in tumor samples: the Catalog of Somatic Mutations in Cancer (COSMIC) SBS7 and SBS11 associated with exposures to ultraviolet light and alkylating agents, respectively (*Alexandrov et al., 2020*; *Harris, 2015*; *Mathieson and Reich, 2017*). To test whether one of these two mutational signatures may be responsible for the observed differences in polymorphism data, we recalculated the C>T/C>A mutation ratio at non-CpG sites after excluding the sequence contexts most affected by SBS7 or SBS11 ('Materials and methods'). While we observe some reduction in the magnitude of non-CpG C>T/C>A ratio in Europeans, the interpopulation differences remain significant (*Figure 2B*). These results suggest the transient change in non-CpG C>T/C>A ratio is not fully driven by the mutational mechanisms corresponding to either COSMIC SBS7 or SBS11. Thus, the etiology of this signal in Europeans remains unclear.

## Divergence of C>G/T>A ratio among populations

The second largest interpopulation difference is in the C>G/T>A ratio (*Figure 2A*) following the OOA migration among all three populations. In the past 3000 generations, both YRI and CEU samples show an increase in the C>G/T>A ratio albeit of different magnitudes, while in the CHB, the ratio initially decreases and then stays relatively stable for roughly 900 generations (*Figure 2A*). Interestingly, unlike the previous signal, interpopulation differences in C>G/T>A remain highly significant for the most recent variants as well (0–55 generations), pointing to ongoing factors differentiating the relative rates of C>G and T>A mutations at present.

The fraction of C>G in de novo germline mutations is particularly sensitive to parental ages, increasing rapidly with the mother's age at conception (*Jónsson et al., 2017*). This raises the possibility that the interpopulation differences in C>G/T>A ratio are driven by different average maternal reproductive ages among populations (*Coll Macià et al., 2021*). To test this hypothesis, we leveraged the regional enrichment of maternal C>G mutations – 'C>G enriched regions' – defined as 10% of the genome with the highest C>G SNP density that contributes to one-third of the overall maternal age effect (i.e., the yearly increase in maternal DNMs with mother's age; *Jónsson et al., 2017*). The C>G/T>A ratio within the C>G enriched regions does not show significant interpopulation differences (*Figure 2C*), possibly reflecting reduced power due to the much lower SNP counts in these regions (<15% of all SNPs; but see *Figure 3—figure supplement 4* for power simulation). Outside of the C>G enriched regions, the three populations differ as much as they do genome-wide (*Figure 2C*), indicating that the differential accumulation of C>G mutations with maternal ages is not the primary driver of the differences observed across these populations.

To determine whether the signal in C>G/T>A ratio is driven by differences between populations in C>G or T>A mutation rate, we performed two additional comparisons (T>G/T>A and C>G/C>A), substituting numerator or denominator in the ratio by another mutation type. Unlike previous comparisons, these two comparisons are sensitive to the effects of gBGC, so the variation across time bins and populations cannot be readily interpreted as evidence for an evolution of the mutation spectrum. However, if the interpopulation differences are in the same direction (i.e., rates in CHB < CEU < YRI), we can reason that the mutation type that is not being substituted (C>G or T>A) contributes to the interpopulation differences. For T>G/T>A ratio, we still observe highly significant interpopulation differences across the three populations, with CEU and YRI converging in recent time windows (*Figure 2—figure supplement 6*). Considering the C>G/C>A ratio, we also find subtle but significant differences during the period of 55–437 generations ago (*Figure 2—figure supplement 6*). These results suggest that the interpopulation differences in the C>G/T>A ratio arise from differences in mutation rates in both numerator and denominator, with CHB having the highest T>A and lowest C>G rates, and YRI having the lowest T>A and highest C>G rates.

## Differences in the T>C/T>G ratios at deep timescales in human evolution

The T>C/T>G ratios are higher in the three oldest bins (dated to >28,800 generations ago) than in more recent ones in all three population samples and the effect is more pronounced in YRI compared to CHB and CEU (*Figure 2A*). The difference between old polymorphisms observed in different contemporary populations is puzzling because the majority of these variants long predate the OOA migration ~2000–4000 generations ago (*Schiffels and Durbin, 2014*), and thus must have arisen in the common ancestor of the three contemporary populations. We also observed a significant though small difference in T>C/T>G ratio between CHB and other populations at recent timescales (*Figure 2A*), but the signal is no longer significant after removing singletons (*Figure 2—figure supplement 13*).

We performed additional analysis to verify that these apparent interpopulation differences are not driven by biases or inaccuracies in dating mutations. For each variant, Relate inferred an initial estimate of mutation age based on the entire 1000 Genomes data across populations and a refined estimate for each population by applying an iterative Markov Chain Monte Carlo (MCMC) algorithm to the data from that population. Depending on the sample size and demographic history of each population, the refined population-specific estimates could be differentially biased or associated with varying degrees of uncertainty, which can produce spurious differences across populations. We find systematic differences in the initial and refined mutation ages, but the population-specific estimates for the same variant found in two populations have overlapping age ranges for over 90% of the

variants (*Figure 2—figure supplement 7*). Moreover, the T>C/T>G signal appears to be primarily driven by non-shared variants across populations (p <10⁻⁸ by chi-square tests on variants in the three oldest bins), and as expected the interpopulation differences are not significant among variants shared by all three populations (p >0.01 by chi-square tests; *Figure 2—figure supplement 8*).

Next, we examined whether the elevated T>C/T>G ratio is related to increased T>C or reduced T>G mutations. Using alternative pairwise comparisons (T>C/T>A and T>A/T>G), we infer that the signal is primarily driven by higher proportions of T>C mutations among older variants (*Figure 2—figure supplement 9*). We then investigated the sequence context in which the T>C signal is enriched by applying non-negative matrix factorization (NMF), which has been extensively used in analysis of somatic and germline mutations to reveal combinations of mutation types caused by the same mutational process (i.e., mutational signatures; *Alexandrov et al., 2020*; *Mathieson and Reich, 2017*; *Seplyarskiy et al., 2021*). Two of the three significant signatures identified by NMF (based on standard diagnostic criteria, 'Materials and methods') show the same trends as the non-CpG C>T/C>A and T>C/T>G ratios we observe. The mutational signature corresponding to the T>C/T>G ratio is characterized by NTG>NCG and ATW>ACW mutations (*Figure 2—figure supplement 10*).

The enrichment of signal in NTG contexts raises the possibility that some of the 'old' TpG>CpG mutations may actually be mis-polarized CpG>TpG mutations that are hyper-mutable and more likely to undergo recurrent mutations in the 1000 Genomes samples. Consistent with this hypothesis, T>C variants dated to the last few age bins (>28,000 generations) contain substantially greater fractions of TpG>CpG mutations relative to those in the younger bins (*Figure 2—figure supplement 11*). Notably, when variants at TpG sites were excluded, both the elevation in T>C/T>G ratio in old variants and the interpopulation differences disappear (*Figure 2D*).

Previous analyses have inferred that the genome-wide rate of mis-polarization of the ancestral alleles is ~1–4% (*Glémin et al., 2015*; *Hernandez et al., 2007a*). To minimize the effect of ancestral misidentification errors on the T>C/T>G ratio, we applied several standard approaches. We repeated the analysis with the ancestral allele inferred (1) using only the high-confidence sites in EPO alignment and (2) using the chimpanzee reference genome (panTro2) that is equally distant to all modern human populations for the inference ('Materials and methods'). In both cases, we find qualitatively similar results to *Figure 2*, though the population differences are more significant in (2) (*Figure 2—figure supplement 12A and B*). Given that the human reference genome is derived from multiple individuals and has European, African, and East Asian ancestry in different regions, we were concerned about the potential impact of reference bias on our results (*Green et al., 2010*). Thus, we stratified the human reference genome by the inferred local ancestry and again, obtained qualitatively similar results in regions of European or African ancestry (*Figure 2—figure supplement 12C*). Together, our analyses suggest that potential mis-polarization errors at hypermutable CpG sites could have a non-negligible impact on the T>C/T>G signal in ancient variants, but these are challenging to properly correct for using standard approaches. Out of caution, we therefore excluded TpG sites from downstream analysis.

## Parental age effects on the mutation spectrum

To explore whether the interpopulation differences in polymorphism data could be driven by changing mean generation times over evolution (*Coll Macià et al., 2021*), we turned to genomic data from present-day pedigrees and quantified the parental age effects on the pairwise ratios of DNMs ('Materials and methods'). To maximize the power and precision, we focused on the largest published DNM dataset, which includes 200,435 DNMs from 2976 Icelandic trios (*Halldorsson et al., 2019*). The inferred parental age effects based on a previous, smaller DNM dataset were qualitatively similar (*Figure 3—figure supplement 1*), despite some significant differences across datasets, possibly due to systematic differences in the criteria for identifying and filtering DNMs (*Figure 3—figure supplement 2*).

Considering the four pairwise mutation ratios, which are mathematically independent, three show a significant dependence on parental age (*Figure 3*). As an illustration, if both parents reproduce at 40 years rather than at 20 years of age, the ratios of non-CpG C>T/C>A and non-TpG T>C/T>G decrease by 9.4% (90% confidence interval [CI]: 4.3–14.6%; *Figure 3A*) and 7.5% (90% CI: 1.0–13.8%; *Figure 3C*), respectively, whereas the C>G/T>A ratio increases by 11.9% (90% CI: 4.1–20.0%; *Figure 3B*). In terms of sex-specific effects (*Figure 3—figure supplement 3*), non-CpG C>T/C>A and

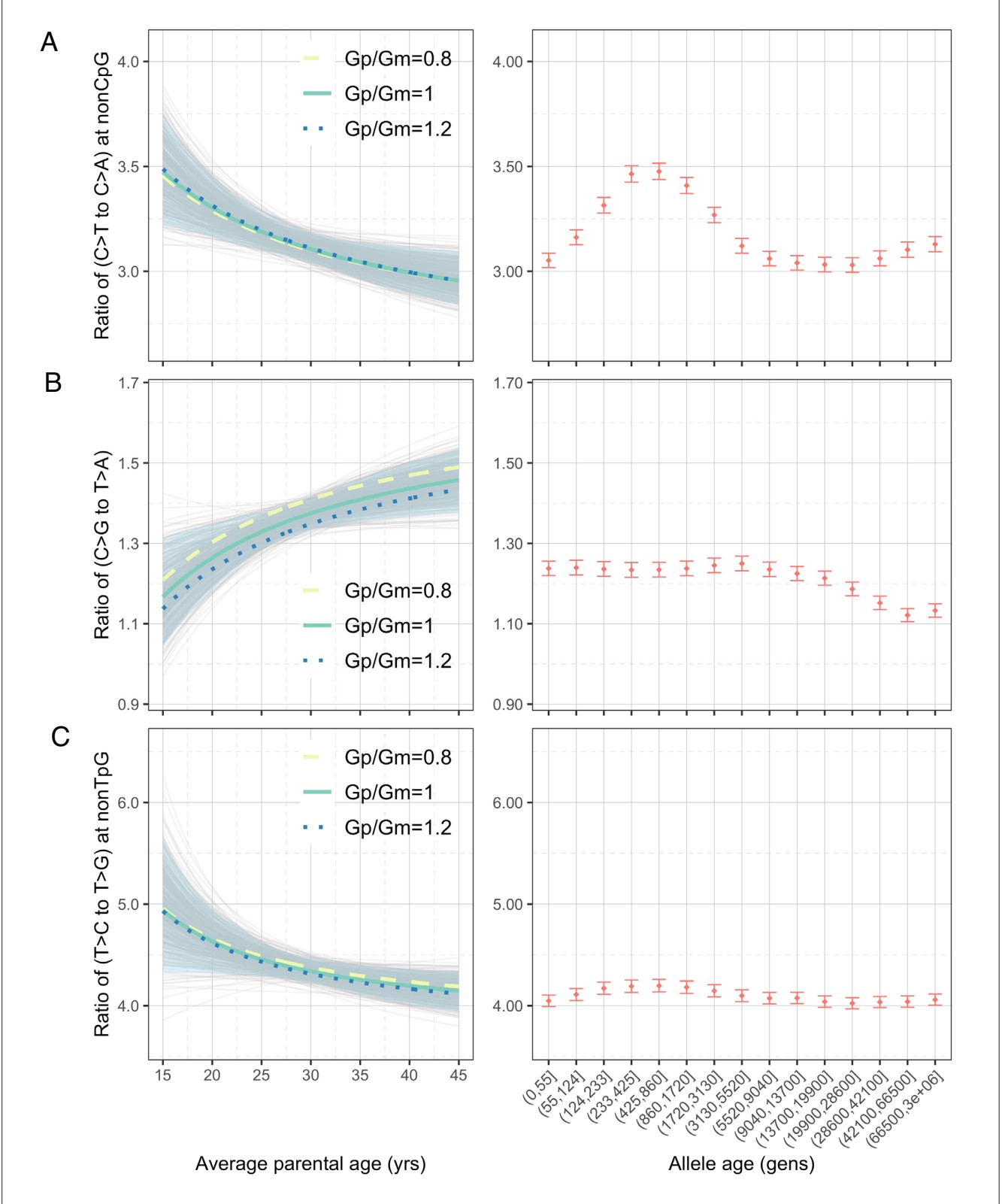

**Figure 3.** Effects of parental ages on three pairwise mutation ratios estimated from de novo mutation (DNM) data in 2879 Icelandic trios (*Halldorsson et al., 2019*). The three panels show the parental age effects (left) on (**A**) nonCpG C>T/C>A, (**B**) C>G/T>A, and (**C**) nonTpG T>C/T>G ratios, respectively. On the left, the different colored curves reflect expected mutation ratios for different ratios of paternal (Gp) to maternal (Gm) mean generation times. Each light gray curve represents the expected ratio for Gp/Gm = 1 from one bootstrap resampling replicate (see 'Materials

*Figure 3 continued on next page*

*Figure 3 continued*

and methods'), with the lighter blue area denoting 90% confidence interval (CI) assessed from 500 replicates. For ease of comparison, ratios for polymorphisms of different ages identified in CEU are shown on the right of each panel. The points represent the observed polymorphism ratios, while the whiskers denote the 95% CI assuming a binomial distribution of polymorphism counts.

The online version of this article includes the following source data and figure supplement(s) for figure 3:

**Source data 1.** Mutation parameters inferred from de novo mutation (DNM) data in 2879 Icelandic trios with estimated uncertainty based on bootstrap resampling (one file for each mutation type for commonly accessible regions; n = 500 replicates).

**Figure supplement 1.** Effects of parental ages on three pairwise mutation ratios estimated from an earlier de novo mutation (DNM) dataset (*Jónsson et al., 2017*).

**Figure supplement 2.** Discrepancies in the mutation spectrum between de novo mutation (DNM) datasets and between DNMs and young polymorphisms.

**Figure supplement 3.** Sex-specific parental age effects on three pairwise mutation ratios.

**Figure supplement 4.** Power for detecting interpopulation differences in C>G/T>A ratio driven by differences in generation time outside (**A**) and within (**B**) the maternal C>G enriched regions.

non-TpG T>C/T>G ratios are largely determined by the paternal age and much less so the maternal age, reflecting that the paternal age effect is three- to fourfold stronger than the maternal age effect for these mutation types (*Goldmann et al., 2018*; *Jónsson et al., 2017*; *Kong et al., 2012*). For C>G/T>A ratio, however, the maternal age is nearly as important as the paternal age, consistent with the unusually strong maternal age effect on C>G mutations (*Jónsson et al., 2017*).

We were unable to directly quantify the dependence of CpG C>T/C>A ratio on parental ages because the low count of C>A mutations at CpGs (on average 0.55 DNMs per trio) limits our ability to reliably infer the parental age effects ('Materials and methods'). However, a previous study noted that the fraction of CpG C>T mutations among all DNMs depends strongly on parental age and decreases by 0.26% per year (*Jónsson et al., 2017*). Consistent with this finding, the ratio of the counts of CpG C>T and CpG C>A DNMs differs significantly: the 20%-tile of trios with the youngest parents have a significantly higher ratio than in the 20%-tile of trios with the oldest parents (21.0 vs. 17.4, p=0.03 by chi-square test). This difference suggests the ratio of C>T to C>A mutations at CpG sites likely decreases with parental age. Overall, the significant age-dependency of three, and likely all, of the four pairwise mutation ratios highlights the pervasive influence of reproductive ages on the human germline mutation spectrum.

## Shifts in generation times needed to explain the observed changes in the polymorphism data

Motivated by the strong dependency of DNM ratios on parental ages, we tested the hypothesis that changes in past generation times account for the observed mutation spectrum of polymorphism data, as suggested by a couple of recent studies (*Coll Macià et al., 2021*; *Wang et al., 2023*). In particular, we asked whether the temporal shifts in the pairwise polymorphism ratios could be fully explained by shifts in average reproductive ages, that is, without the need to invoke additional factors. As the mutation process may have evolved possibly separately in different human populations, we focused the comparison of DNMs to variants identified in CEU, who are most genetically similar to the Icelandic individuals (with $F_{ST}$<0.005) for whom we have the largest DNM dataset (*Halldorsson et al., 2019*).

Assuming the observed changes in the mutation spectrum are solely driven by shifts in average reproductive ages, we inferred past generation times by relating observed pairwise ratios in DNM data and polymorphism data. Specifically, for a given pairwise mutation ratio in the polymorphism data, we asked what value of the generation time is compatible with the relationships to age estimated from pedigree data (assuming identical male to female mean generation times and a fixed onset of puberty). Accounting for uncertainty in the DNM data, we then inferred the 95% confidence interval of the generation time for each mutation ratio and time window. Given the complications with low numbers of CpG mutations in DNMs data and of recurrent mutations at CpG sites in polymorphism data, we excluded the pairwise ratio of C>T/C>A at CpGs for this analysis.

We inferred the generation times across mutation ages in CEU by solving for the parental ages that would give rise to the observed pairwise ratios in each time window. Surprisingly, the estimates are inconsistent for different mutation ratios within a single mutation age bin. Moreover, the overall

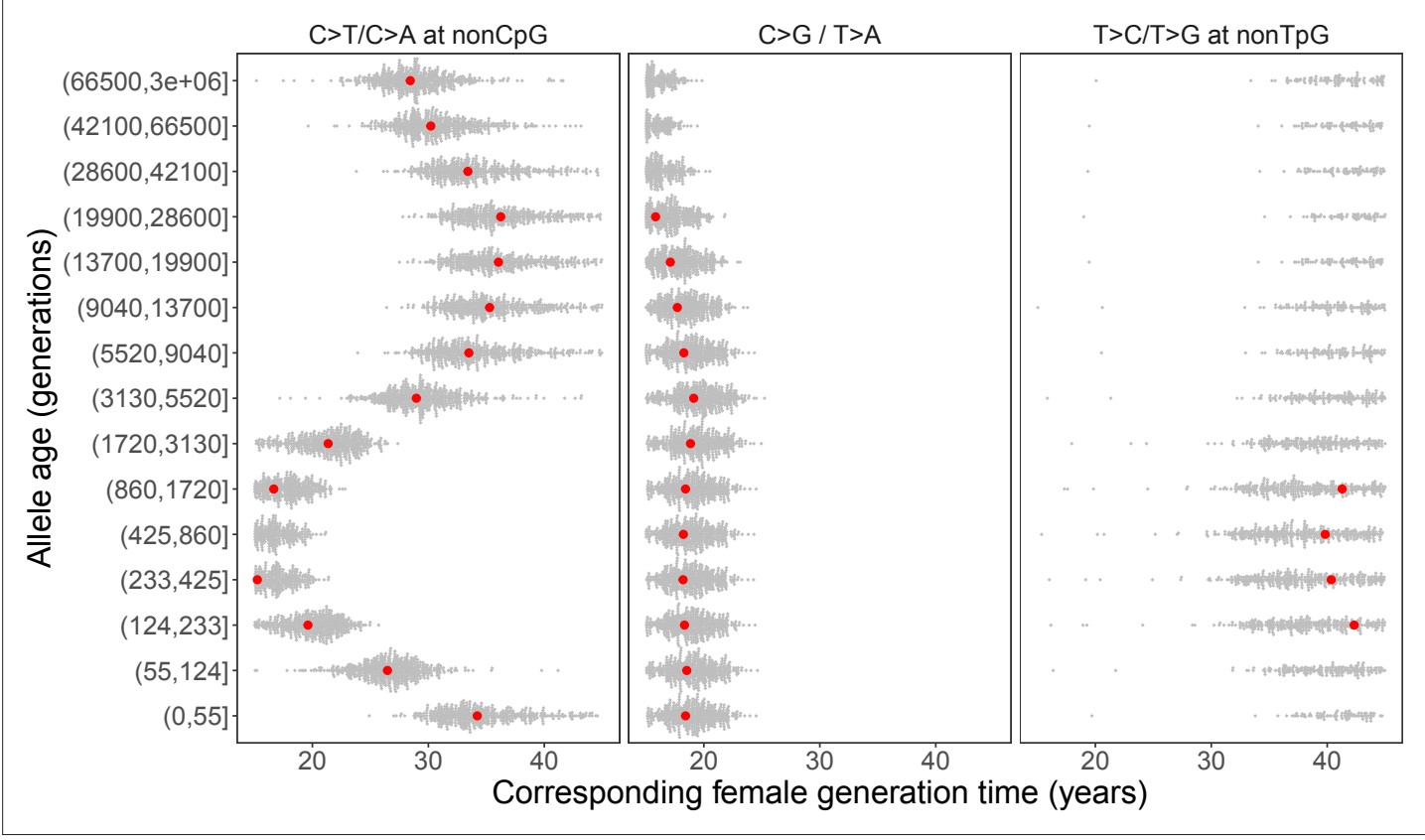

**Figure 4.** Past generation times corresponding to the observed polymorphism ratios in CEU, given parental age effects estimated from de novo mutation (DNM) data. Red points represent the point estimates based on maximum likelihood estimators of mutation parameters from the DNM data; gray dots show estimates from 500 bootstrap replicates by resampling trios with replacement. We assumed the same male to female generation times (Gp = Gm) for all time windows. Similar trends were obtained for other fixed values of Gp/Gm (between 0.8–1.2) or independently varying Gp and Gm (*Figure 4—figure supplement 1*, *Figure 4—figure supplement 3*).

The online version of this article includes the following source data and figure supplement(s) for figure 4:

**Source data 1.** Past generation times inferred from each polymorphism ratio, assuming fixed ratios of male to female generation times (Gp/Gm = 0.8, 1, 1.1, 1.2), with confidence intervals estimated using bootstrap resampling (n = 500 replicates; one file for each mutation type).

**Figure supplement 1.** Past generation times corresponding to the observed pairwise polymorphism ratios, assuming fixed ratio of male to female generation times of 0.8 (**A**), 1.1 (**B**), and 1.2 (**C**).

**Figure supplement 2.** Past generation times corresponding to the observed pairwise polymorphism ratios, based on parental age effects estimated from an earlier de novo mutation (DNM) data (*Jónsson et al., 2017*).

**Figure supplement 3.** Combinations of paternal and maternal reproductive ages corresponding to the observed pairwise polymorphism ratios.

trends are inconsistent across time windows. For instance, the steady increase in C>G/T>A ratio over time translates into a gradual increase in reproductive age, with the ratio of the most recent bin corresponding to a reproductive age under 23 years (*Figure 3C*, *Figure 4*). In contrast, the non-TpG T>C/T>G polymorphism ratio appears to be lower than the ratio in DNMs across the range of typical parental ages in pedigree studies and suggests a generation time of more than 40 years (*Figure 3C*). Such a long population-average generation time is not only inconsistent with the estimate of the C>G/T>A ratio, but it is also unrealistic for human evolution (*Fenner, 2005*; *Moorjani et al., 2016b*). Finally, the transient elevation in non-CpG C>T/C>A ratio suggests a drastic, rapid reduction in the generation time (*Figure 3A*, *Figure 4*). Specifically, the ratios of both ancient (>5670 generations ago) and the most recent polymorphisms (<55 generations ago) correspond to relatively old reproductive ages of ~35 years, while the peak at around 238–887 generations ago provides average reproductive age estimates of less than 20 years (*Figure 4*).

The incompatible patterns across different mutation ratios can potentially arise if male and female generation times differed in the past. To explore this possibility, we varied the ratio of male and female generation times between 0.8–1.2, as seen across a range of contemporary human populations (*Fenner, 2005*; *Figure 4—figure supplement 1*, *Figure 4—figure supplement 2*). We further allowed male and female generation times to vary freely and inferred the combinations of paternal and maternal ages that could give rise to the observed polymorphism ratios. Even after modeling sex-specific reproductive ages, we find inconsistent generation time estimates and incompatible trends across mutation ratios (*Figure 4—figure supplement 3*).

In addition to the temporal changes within CEU, the observed patterns in mutation spectra across populations cannot be explained by differences in generation times. In particular, given the extraordinarily strong maternal age effect on C>G mutations in maternal hotspots (*Jónsson et al., 2017*), we would expect more pronounced differences in the C>G/T>A ratio in these regions than in the rest of the genome, should the mutation patterns be driven by generation time. Simulations based on the parental age effects observed in pedigree data suggest that the power to detect differences in C>G/T>A ratio due to change in generation times is much greater within the maternal C>G mutation hotspots than in the rest of the genome, despite the much smaller number SNPs in the hotspots (*Figure 3—figure supplement 4*). Therefore, if the interpopulation differences in C>G/T>A ratio were solely driven by differences in maternal age across populations, we should be well-powered to observe a signal in the maternal hotspot regions alone. Not seeing one (*Figure 2C*) provides another line of evidence against the premise that generation time is a major driver of the observed mutation patterns across time or between populations in humans.

Together, our results suggest that shifts within a plausible range of human generation times – that is, those that fall between the ages of puberty and reproductive cessation in contemporary humans –cannot explain the observed variation in the polymorphism data for CEU, and by extension are unlikely to explain the mutation ratios in CHB or YRI polymorphisms.

## Discussion

### Multiple changes in the germline mutation spectrum during the course of human evolution

We introduce a new framework to compare the mutation spectrum over time and across population samples, while controlling for the effects of selection and biased gene conversion. By applying this approach to multiple population samples from the 1000 Genomes Project dataset, we observe multiple independent signals of interpopulation differences. Notably, we replicate the transient elevation in non-CpG C>T mutation (manifest in the C>T/C>A ratio) identified previously in Europeans compared to East Asians and Africans (*DeWitt et al., 2021*; *Harris, 2015*; *Harris and Pritchard, 2017*; *Mathieson and Reich, 2017*; *Speidel et al., 2019*). We find that this ratio also differs subtly between YRI and CHB, suggesting an additional change occurred in this mutation type. In both cases, the signal is enriched in the TCC, TCT, CCC, and ACC contexts (*Figure 2—figure supplement 11*), and mutation types that are associated with exposure to UV and alkylating agents (*Harris, 2015*; *Mathieson and Reich, 2017*). While these contexts may contribute to this signal, we show that they do not fully explain the observed differences between contemporary populations. Thus, the etiology of this signal remains obscure, and it may not be specific to groups with west Eurasian ancestry. Further investigation into the extended sequence contexts of this mutation pulse may help elucidate the underlying molecular mechanism(s) (*Aggarwala and Voight, 2016*; *Aikens et al., 2019*).

We also observe two additional interpopulation differences in the mutation spectrum. First, the ratio of C>G/T>A mutation rates differs between YRI, CEU, and CHB. We find that this signal is related to an increase in T>A mutations and depletion of C>G mutations in CHB compared to CEU, as well as a depletion of T>A mutations in YRI. Some aspects of this observation (e.g., the enrichment of T>A mutations in East Asians) were previously noted (*Harris and Pritchard, 2017*), but our analysis adds information about the changes in additional populations and insights into the timing of this change. Given the distinct trends of T>A and C>G mutations with allele age (*Figure 2—figure supplement 6*), it appears that at least two changes in the mutational processes are needed to explain the interpopulation differences. By comparing the ratios inside and outside C>G hotspots, we confidently rule out a primary role of maternal generation times in driving these differences. Interestingly, these differences

are still observed in the most recent polymorphisms, indicating that – unlike the TCC mutation pulse – this process is likely ongoing. This finding therefore points to an opportunity to directly examine and map the underlying biological causes using large-scale DNM datasets from diverse populations.

We also detect a shift in the ratio of T>C/T>G mutations at old timescales (>28,000 generations) and differences among human populations. This signal is driven by T>C rather than T>G and enriched in TpG contexts; in fact, the signal disappears after excluding TpG sites. We hypothesize that some of the inferred ancient TpG>CpG mutations may be mis-polarized CpG>TpG mutations that are hyper-mutable and more likely to undergo recurrent mutations over the course of primate evolution. While there can be evolution in T>C mutation rate at old timescales, controlling for the effects of mis-polarization errors is challenging and hence this signal remains tentative.

## Effects of ancestral mis-polarization in the study of mutation spectrum

Misidentification of the ancestral alleles has been found to be more likely at sites with higher mutation rates and shown to substantially impact studies of natural selection and biased gene conversion (*Eyre-Walker, 1998*; *Glémin et al., 2015*; *Hernandez et al., 2007a*; *Hernandez et al., 2007b*). Accounting for ancestral misidentification is also critical in studies investigating variation in mutation rate and spectrum. Our analysis shows that current methods – using multispecies alignment or tree-based polarization approaches – still suffer from several limitations. We discuss some of these in turn.

For most of the analyses, we used the ancestral allele inferred by six primate EPO alignment. The EPO alignment relies on a model of continuous time DNA nucleotide substitutions. By default, it uses the HKY model, with the ratio of transitions to transversions set to 2, and a stationary GC frequency set to 40% (*Paten et al., 2008*). It does not take into account context-dependent, fine-scale variation in mutation rate, however, these can lead to higher error rate in assigning ancestral alleles for the hyper-mutable CpG sites (*Harpak et al., 2016*). Moreover, it considers only a single reference genome in each species; by not taking into account information about allele frequency of variants within a species, it provides less accurate inference of ancestral alleles (*Hernandez et al., 2007a*). As an alternative to the use of EPO, we used the chimpanzee reference genome to infer the ancestral state and obtained qualitatively similar results, likely because this approach is sensitive to similar biases, in particular at fast evolving sites.

In turn, the program Relate requires ancestral alleles as input and aims to identify misclassified ancestral alleles based on whether the derived allele maps to a unique branch of the gene genealogy. It assumes an infinite-site mutation model according to which all haplotypes carrying the derived allele should be descendants of a unique branch; if this is true for the ancestral allele but not the derived one, Relate 'flips' their assignments (the mutation is left 'unmapped' if neither derived nor ancestral allele can be mapped to a unique branch). In this approach, mis-polarization of the ancestral allele still remains an issue, especially for sites that experience recurrent mutations in the sample. Because Relate assumes the mutation occurred exactly once in the genealogical history of a sample, the placement of mutations is bound to be inaccurate for repeat mutations that need to be mapped to multiple branches of the tree. This issue disproportionately impacts CpG transitions, as exemplified by the fact that CpG>TpG mutations are under-represented in the 'mapped' subset of mutations inferred by Relate (*Speidel et al., 2019*). Explicit modeling of polarization errors has been shown to be effective in reducing bias in the inference of evolutionary parameters using site frequency spectrum data (*Glémin et al., 2015*; *Hernandez et al., 2007b*). However, the unmapped mutations in Relate output are likely unevenly distributed across allele frequency and mutation age, possibly in a demography-dependent manner, making it hard to predict or correct for this effect. Multiple independent CpG>TpG mutations at the same locus on different branches of the gene genealogy may be misclassified as a single old TpG>CpG mutation (and the allele age inferred would also be incorrect). Beyond CpGs, these effects are likely to impact other highly mutable sites (e.g., certain types of transitions) and may spuriously appear as signals of changes in the mutation spectrum.

## Changes in generation times cannot explain the evolution of the mutation spectrum in humans

Across mammals, generation time is the strongest predictor of the yearly mutation rate and of some aspects of the mutation spectrum (*Hwang and Green, 2004*; *Moorjani et al., 2016a*; *Wu and Li, 1985*). Accordingly, our analysis of DNMs from pedigree studies shows significant effects of parental

ages on all four pairwise mutation ratios that we examined. Recent studies have argued that changes in generation times can explain a large fraction of the differences in mutation spectrum observed across human populations (*Coll Macià et al., 2021*), and one study even leveraged the population-specific mutation spectrum to infer the historical generation time in humans (*Wang et al., 2023*).

It may seem like a promising idea to infer generation time based on changes in mutation spectrum, but in practice, several technical hurdles stand in the way. First, given the sampling noise associated with the limited number of DNMs per family, large numbers of pedigrees are required to characterize the parental age effects reliably, especially for very specific mutation types. In our analysis, we used the largest available pedigree dataset, but the parental age effects remain imprecisely estimated and should be revisited as larger datasets become available, ideally from a diverse set of populations. Second, technical issues, both molecular and computational, may affect the reliability of variant calls of different mutation types (*Bergeron et al., 2021*). Indeed, we find that the four pairwise ratios differ significantly across two recent pedigree datasets, as well as between DNMs and young polymorphisms in 1000 Genomes dataset, which are unlikely to be reconciled by biological reasons (*Figure 3—figure supplement 2*). In addition, controlling for the effects of biased gene conversion is difficult, as its effects may differ to some extent by mutation type even within a class (for instance, T>Cand T>Gboth classified as S>W mutations may be subject to sublte differences in strengths of BGC).

Importantly, fitting a generation time to observed differences in mutation spectrum relies on the assumption that changes in the generation time play the sole (or at least the predominant) role in the evolution of the mutation spectrum, an assumption that does not seem to hold in data. Specifically, we find that the generation time estimates inferred from different mutation ratios independently disagree with each other, within the same time window or over time. Notably, the temporal trends, which should be robust to most technical issues mentioned above, inferred from different mutation ratios are mutually inconsistent. In other words, changes in a single parameter – generation time – cannot explain the mutation patterns in humans (*Figure 4*). These inconsistencies persist after accounting for uncertainty in the parental age effects inferred from pedigrees and incorporating sex-specific reproductive ages. Moreover, the significant divergence in C>G/T>A ratio outside the maternal C>G hotspots but not within the hotspots argues against parental age (in particular, maternal age) as a major driver of mutation spectrum differences across populations (*Figure 3—figure supplement 4*). These findings thus establish that changes in generation time alone cannot account for all or even most of the observed variation in mutation spectrum over the course of human evolution.

## Implications

The mutation spectrum of polymorphisms is a convolution of multiple evolutionary forces: mutation, recombination (including gene conversion), natural selection, and their interplay with demography. In this study, we investigated the contribution of these forces to differences in the mutation spectrum across contemporary human populations. For future studies aiming to understand the evolution of mutagenesis based on analyses of polymorphism patterns, it will be crucial to consider more realistic mutation models (including using context-dependent models for ancestral allele reconstruction) and account for the impact of non-mutational evolutionary forces.

Our analysis demonstrates the limitations of inferring past generation times based on polymorphism patterns. We find that shifts in generation time alone cannot explain the observed variation in the mutation spectrum, leaving a non-negligible role for other factors – such as transient environmental exposures, genetic modifiers and other life history traits (e.g., changes in the onset of puberty) – in shaping the mutation landscape in human populations. This conclusion is in line with recent studies in model organisms that discovered naturally occurring genetic modifiers (*Jiang et al., 2021*; *Sasani et al., 2022*) as well as a human pedigree study that identified individuals with germline hypermutation potentially due to genetic modifiers or exposures to chemotherapeutic agents (*Kaplanis et al., 2021*).

Although the three population samples that we focused on here were collected from three distinct continents, the observed differences among them are not necessarily generalizable to continental level. In particular, within the same continent, and notably in Africa, there is relatively deep genetic divergence between some populations (e.g., between Bantu groups and Khoe-San), often accompanied by long-term geographic isolation and environmental differences (*Mallick et al., 2016*). These different histories can lead to considerable variation in the mutation processes within a continent. In

fact, even for closely related populations, we detect subtle but significant differences in the polymorphism ratios (e.g., between CEU and TSI in *Figure 2—figure supplement 1*). Genetic data from more diverse populations, in terms of both ancestry and geographic location, are needed to generate a more complete picture of past and ongoing variation in the mutation spectrum across human ancestries and to understand its evolution.

The variation in the mutation spectrum over the course of human evolution raises a fundamental puzzle about why the molecular clock works over long timescales and across species. Our analyses uncovered substantial variation in multiple pairwise mutation ratios at different time depths during human evolution. Since variation in each pairwise ratio suggests mutation rate variation for at least one (or both) of the mutation types involved, our findings suggest that the absolute mutation rate per year of several mutation types must have been evolving. For example, our result suggests the mutation rate for C>T mutations at non-CpG sites varied by ~15–20% in CEU over the past 3000 generations. Over longer timescales, it is likely that all mutation types deviate from a strictly clock-like behavior. It is puzzling then that the mutation rates across species are strikingly similar over millions of years: for instance, the substitution rates differ by less than 10% for any mutation type across the human and chimpanzee lineages (*Moorjani et al., 2016a*). This observation suggests that although the mutation rate and spectrum can evolve over relatively short timescales, the fluctuations in yearly mutation rate often average out over longer timescales, possibly reflecting the effects of long-term stabilizing selection.

## Materials and methods

| Dataset | Source | Reference |
|---|---|---|
| High coverage 1000 Genomes Project | https://www.internationalgenome.org/data-portal/data-collection/30x-grch38 | *Byrska-Bishop et al., 2022* |
| Decode de novo mutations 2019 (2976 families) | https://www.science.org/doi/10.1126/science.aau1043 | *Halldorsson et al., 2019* |
| Decode de novo mutations 2017 (1548 families) | https://www.nature.com/articles/nature24018#additional-information | *Jónsson et al., 2017* |

### Data filtering and partitions used in the analysis

#### Commonly accessible regions

In order to reliably compare mutation patterns across datasets, we generated a list of genomic regions that were 'accessible' or assayed by the study after accounting for the constraints of the study design. To generate this list, we first followed the variant calling procedure described in *Jónsson et al., 2017* to identify the accessible genome for de novo studies. This yielded an accessible genome length of 2.7 Gb similar to the estimate reported in the original study (*Jónsson et al., 2017*). We intersected this dataset with the 1000 Genomes Strict Mask (see 'Resources' below). We used the strict mask generated using low-coverage 1000 Genomes dataset as it encompasses a larger set of low-complexity regions and thus may port well across datasets. Further, to focus on putatively neutral regions, we removed exons and phylogenetically conserved regions sources. The combined set of accessible autosomal regions contained 2.15 Gb. Unless otherwise stated, we present all results generated for this subset of the autosomal genome, which we refer to as the 'commonly accessible' regions.

#### Regions of high and low recombination rate

To study the impact of recombination rate on mutation patterns, we divided the genome into bins sorted by recombination rate using the HapMap recombination map (*1000 Genomes Project Consortium et al., 2015*). We then sorted all genomic sites by recombination rate and divided the genome into three discrete bins with recombination rates of (0, 0.0717), (0.0717, 0.422), and (0.422, ∞) cM/Mb, each containing roughly 33% of the genomic bases.

## B-statistic or B-scores

To focus on regions of the genome that are minimally affected by linked selection, we assigned a *B*-score to each variant site in a population. The *B*-score measures the expected reduction in diversity levels at a site due to background selection, with smaller values implying greater effects of background selection. We used the *B*-score values provided by *McVicker et al., 2009*. We then compared the mutation patterns within windows of different values of B-scores. Additionally, where specified, we used the list of effectively neutral regions that contains the commonly accessible regions with a B-score >800. This subset of the genome includes 1.33 Gb.

## Human ancestral allele reconstruction

For most analyses, we used the ancestral allele reconstruction based on the six primate EPO alignment (*1000 Genomes Project Consortium et al., 2015*). In the EPO (Enredo-Pecan-Ortheus) pipeline, Ortheus infers ancestral states from the Pecan alignments (*Paten et al., 2008*). The confidence in the ancestral call is determined by comparing the call to the ancestor of the ancestral sequence as well as the 'sister' sequence of the query species. High-confidence sites are annotated with capital letters in the alignment. For some analysis, we also used the chimpanzee (panTro2) reference genome to infer the ancestral allele (mapped to human reference genome coordinates, *hg19*).

## Resources

| Dataset | Source link |
| --- | --- |
| 1000 Genomes Strict Mask | https://www.internationalgenome.org/announcements/genome-accessibility-masks/ |
| Recombination rate map | https://alkesgroup.broadinstitute.org/Eagle/downloads/tables/genetic_map_hg38_withX.txt.gz |
| B-scores | https://journals.plos.org/plosgenetics/article?id=10.1371/journal.pgen.1000471 |
| Conserved regions | http://hgdownload.cse.ucsc.edu/goldenPath/hg38/database/phastConsElements46wayPrimates.txt.gz |
| Coding regions | http://hgdownload.cse.ucsc.edu/goldenPath/hg38/database/refGene.txt.gz |
| Human ancestral genome | https://ftp.ensembl.org/pub/release-86/fasta/ancestral_alleles/homo_sapiens_ancestor_GRCh38_e86.tar.gz |
| Chimpanzee reference genome (panTro2) in hg19 coordinates | https://reichdata.hms.harvard.edu/pub/datasets/sgdp/ |
| COSMIC signatures (v3.2, GRCh38) | https://cancer.sanger.ac.uk/signatures/documents/453/COSMIC_v3.2_SBS_GRCh38.txt |

## Relate analysis

We applied Relate v1.1.5 (*Speidel et al., 2019*) to phased whole-genome sequences from the 1000 Genomes Project (see 'Datasets'). The 1000 Genomes data was phased and imputed statistically (*Byrska-Bishop et al., 2022*). As a result, most singletons, which are missing in the phased data, were not included in Relate analysis. We focused on biallelic SNPs only using VCFtools (`--remove-indels --min-alleles 2 --max-alleles 2`) (*Danecek et al., 2011*). We then converted VCFs to haps/sample format using RelateFileFormats (--mode ConvertFromVcf) and prepared the input files using PrepareInputFiles.sh provided by Relate. We used 1000 Genomes Pilot Mask as the genome accessibility filter and polarized each allele to ancestral or derived state using the six primate EPO alignment. We assumed a mutation rate (*m*) of $1.25 \times 10^{-8}$ per base pair per generation and an effective population size (*N*) of 30,000 (*Jónsson et al., 2017*). We used the HapMap II genetic map (*1000 Genomes Project Consortium et al., 2015*). We first inferred the mutation ages using the entire dataset. Following Relate's guidelines, we inferred a refined estimate for each population by splitting the Relate output genealogies into subtrees using RelateExtract (--mode SubTreesForSubpopulation) and re-estimated the branch lengths (using EstimatePopulationSize.sh) to obtain the final mutation ages and the associated uncertainty (upper and lower bounds that reflect the start and end points

of the branch that the mutation falls on in the reconstructed genealogical tree). For each mutation, we then inferred the upstream and downstream base pair using the six primate EPO alignment. We excluded sites where either the upstream or downstream base in the human ancestral genome was missing or ambiguous. Unless otherwise stated, we present the results for the commonly accessible regions.

Using chimpanzee reference genome to infer the ancestral allele: To test the robustness of our results to the inferred ancestral allele, we reran the Relate analysis with the chimpanzee (panTro2) reference genome mapped to the human reference genome (hg19). Because we only had access to the chimpanzee genome in hg19 coordinates, we first lifted over the vcf files of 1000 Genomes high-coverage data from hg38 to hg19 coordinates using *CrossMap* (*Zhao et al., 2014*). We then polarized the ancestral and derived alleles with respect to the allele in the chimpanzee reference genome and used the HapMap II genetic map in hg19 coordinates (*1000 Genomes Project Consortium et al., 2015*). The rest of the parameters and setup in Relate was identical as previously described.

We note that for all the analyses reported in the article, we focused on SNPs for which Relate successfully inferred the allele ages. Thus, unphased, multiallelic and unmapped SNPs were excluded. This likely introduces some biases in comparison of mutation spectrum especially for young variants, so the Relate results of very recent bins should be interpreted with caution.

## Classification of shared and nonshared variants across continental groups

Considering the six populations under study (YRI, LWK, CEU, TSI, CHB, and JPT), we operationally defined 'shared variants' as SNPs in Relate outputs that have both alleles observed in samples from at least one population from each of the three continental groups, that is, variants segregating in (YRI or LWK) and (CEU or TSI) and (CHB or JPT). Conversely, SNPs that do not meet the above criteria were classified as 'non-shared variants,' although many of them may actually be shared by populations from two (but not three) of the continental groups.

## Binning of polymorphisms based on mutation age

Among SNPs with Relate-inferred allele ages, we filtered out those with extremely old ages (i.e., upper bounds of allele age greater than 3,000,000 generations), as those were too old to be compatible with reasonable human evolutionary history. There is large uncertainty in the mutation ages estimated by Relate, such that the estimated lower and upper bounds often differ by an order of magnitude or more. We took a two-step approach to bin the polymorphisms by age, accounting for this uncertainty. First, we determined the boundaries of age bins by sorting all SNPs segregating in CEU into 15 bins of roughly equal sizes with a Monte Carlo method (i.e., randomly selecting a point estimate by sampling a point uniformly between the upper and lower bounds of inferred allele age by Relate). We then calculated the *pseudo-counts* of each mutation type in each bin by summing up the probability densities across all variants, assuming a uniform distribution of each variant within the inferred age intervals. For example, if a T>A SNP has an estimated age range of (500, 1300) generations, which overlaps with three of the predetermined age bins (312, 545), (545, 1160), and (1160, 2970), we would assign the T>A SNP to three bins with the following weights (545-500)/(1300–800) = 0.056, (1160–545)/(1300–800) = 0.769, and (1300–1160)/(1300–800) = 0.175, respectively. We note that since the allele age distribution differs across populations due to differences in their demographic history, there is no way to bin variants equally for all populations simultaneously. For results shown in main figures, we based our binning into equal sizes on the age estimates of variants observed in CEU. Results were qualitatively similar when the bin boundaries were determined based on variants observed in YRI and CHB samples (*Figure 1—figure supplement 8*, *Figure 2—figure supplement 5*).

## Calculating confidence intervals of polymorphism ratios and the statistical significance of interpopulation differences

To assess the confidence intervals (CIs) of the mutation ratios in polymorphism data, we assumed the pseudo-counts of the two mutation types being compared follow a binomial distribution conditional on the total count. In practice, we used the normal approximation for calculating the 95% CI of the proportion for a given mutation type based on the observed counts of two types, using $\hat{p} \pm z\sqrt{\frac{\hat{p}(1-\hat{p})}{n_1+n_2}}$, where $n_1$ and $n_2$ are the pseudo-counts of two mutation types, $\hat{p} = \frac{n_1}{n_1+n_2}$ is the point estimate of the

probability of success, and $z = 1.96$ is the $Z$-score corresponding to the upper 2.5%-tile. We then transformed the CI of fraction of one mutation type into that of the ratio of the two mutation types using $\left(\alpha_{lower}, \alpha_{upper}\right) = \left(\frac{p_{lower}}{1-p_{lower}}, \frac{p_{upper}}{1-p_{upper}}\right)$.

We performed chi-square tests to evaluate the statistical significance of interpopulation differences in observed mutation ratios. Specifically, for each mutation ratio in each age bin, we constructed a 2 × $N_{pop}$ contingency table, where each entry is the pseudo-count of observed polymorphisms of one of the two mutation types in a population. We then calculated the p-value of the $\chi^2$ statistic and corrected for multiple hypothesis testing by Bonferroni correction by multiplying the p-value by 15 × 4, which represents the product of the number of age bins and the number of mutation ratios studied (for *Figure 2—figure supplement 4*, we substituted the first number by the number of derived allele frequency bins).

## Non-negative matrix factorization (NMF) analysis

To investigate the sequence context in which the T>C signal is enriched, we applied NMF analysis. We applied NMF to the 96 × 45 dimensions matrix containing the normalized allele counts for 96 mutation types (considering the flanking 5′ and 3′ base nucleotides neighboring each SNP) for 15 mutation age bins in the three populations (CEU, CHB, and YRI); normalization was done by dividing the count of alleles by the number of SNPs in each age bin (i.e., within each column). We applied NMF using the R package *MutationalPatterns* and the *brunet* algorithm (*Brunet et al., 2004*; *Manders et al., 2022*) with factorization ranks of 2–15. We chose the factorization rank of three as it explains >99% variance and has the highest cophenetic correlation coefficient (which starts decreasing after $K > 3$) (*Figure 2—figure supplement 10*). Two of the three signatures identified by NMF align well with interpopulation differences identified in our pairwise ratio analysis (*Figure 2*). In particular, signature 1 corresponds to the non-CpG C>T/C>A, signature 2 corresponds to the T>C/T>G ratios, and signature 3 is a mirror image of signatures 1 and 2 likely due to the constraint imposed by performing the analysis using fractions of allele counts and using a small $K$ value for the analysis (*Figure 2—figure supplement 10*). The NMF signatures 1 and 2 are robust to removal of CpG sites and/or singletons.

## Sequence contexts related to COSMIC mutational signatures SBS7 and SBS11

We downloaded loadings of the single base substitution (SBS) reference signatures on the 96 trinucleotide mutation types from COSMIC website (v3.2, GRCh38; link provided under 'Resources'). We found that both SBS7a/b and SBS11 consist of nearly exclusive C>T mutations, with 86.7% mutations caused by SBS7a/b concentrated in YCN contexts while 70.0% SBS11 mutations are in NCY contexts, where Y represents pyrimidine (i.e., C or T) and N represents any base. Therefore, as proxies for mutations potentially affected by SBS7a/b and SBS11, we removed C>T mutations in YCN and NCY contexts in analysis corresponding to *Figure 2B*.

## Quantification of parental age effects on DNM counts and ratios

We used a model-based approach to quantify the effects of paternal and maternal ages jointly by leveraging information from all phased and unphased DNMs. In short, as described in *Gao et al., 2019*, we modeled the expected number of DNMs inherited from a parent as a linear function of parental age at conception, and assumed that the observed number of DNMs follows a Poisson distribution. Using a maximum likelihood approach, we estimated the sex-specific slopes and intercepts (at age zero) for each mutation ratio. Confidence intervals of the slopes and intercepts were assessed by bootstrap resampling of trios. With these estimated parental age effects, we then predicted the expected count of each mutation type and the pairwise ratios under given combinations of maternal and paternal ages, such as shown in the left panel of *Figure 3*.

For analysis corresponding to *Figures 3 and 4*, we inferred the parental age effects based on a DNM dataset from 2976 Icelandic trios (*Halldorsson et al., 2019*). Five trios have exceedingly large numbers of DNMs given the parental ages (Proband IDs: 24496, 71657, 8008, 64783, 126025) and were removed in our analysis. Given the evidence for a nonlinear effect of maternal age (i.e., a more rapid increase in maternal mutations at older ages) (*Gao et al., 2019*), we further excluded 92 trios with maternal ages above 40 in our analysis. Overall, DNM data from 2879 trios were used for inference of (linear) parental age effects on DNMs. We also replicated the analysis (*Figure 3—figure*

*supplement 1*, *Figure 4—figure supplement 2*) with an earlier dataset of DNMs from 1548 Icelandic trios, excluding 73 trios with maternal ages above 40 (*Jónsson et al., 2017*).

### Power simulations for C>G mutations

We performed simulations to estimate the power of detecting significant differences in the C>G/T>A ratio within and outside the maternal C>G mutation hotspots, assuming that all differences in the mutation spectrum are driven by parental ages. Specifically, we assumed two populations with different generation times (G = 20, 25, 30, 35, 40) and predicted the expected fractions of C>G and T>A mutations among all de novo single-nucleotide mutations, using point estimates of parental age effects estimated as described above. Then, we simulated the numbers of C>G and T>A SNPs assuming two independent binomial distributions with the expected fractions and the observed number of SNPs in CEU (taking the largest number of the 15 age bins). We then applied chi-square test on the 2 × 2 contingency table (two mutation types × two populations) to evaluate significance of interpopulation difference. We performed this simulation 10,000 times for the maternal C>G mutation hotspots and other regions separately and estimated the detection power as the fraction of replicates with p<0.001 by chi-square test in the two regions (*Figure 3—figure supplement 4*).

### Inference of generation time corresponding to the observed polymorphism ratios

Under the scenario of co-varying paternal and maternal reproductive ages, we inferred the generation time by solving the system of linear equations:

$\frac{Gp}{Gm} = \gamma$, where $\gamma$ = 0.8, 1, 1.1, or 1.2 is the assumed ratio of paternal to maternal ages; and $\frac{\left(\beta_p^1 Gp + \alpha_p^1\right) + \left(\beta_m^1 Gm + \alpha_m^1\right)}{\left(\beta_p^2 Gp + \alpha_p^2\right) + \left(\beta_m^2 Gm + \alpha_m^2\right)} = R_{1,2}$ , where $\beta$ and $\alpha$ are the slopes and intercepts estimated from DNM data for maternal (*m*) or paternal age (*p*) effects and *R* is the observed ratio of pseudo-counts of two mutation types (indicated with superscript 1,2) in an age bin.

To evaluate the uncertainty in the generation time estimates, we solved the equation system with maximum likelihood estimates from each bootstrap replicate of pedigree data and obtained 90% CIs of the inferred generation times from the overall distribution of estimates across all replicates.

Under the scenario of independently varying paternal and maternal reproductive ages, the combinations of (*Gp*, *Gm*) that satisfy $\frac{\left(\beta_p^1 Gp + \alpha_p^1\right) + \left(\beta_m^1 Gm + \alpha_m^1\right)}{\left(\beta_p^2 Gp + \alpha_p^2\right) + \left(\beta_m^2 Gm + \alpha_m^2\right)} = R_{1,2}$ follow a simple linear constraint, when other parameters are set. Therefore, we plotted in a two-dimensional contour plot the linear combinations of (*Gp*, *Gm*) corresponding to each observed polymorphism ratio from both the maximum likelihood estimators of mutation parameters and all bootstrap estimates (*Figure 4—figure supplement 3*). We then compared the distribution of linear constraints across mutation ratios (*Figure 4—figure supplement 3*).

## Acknowledgements

We thank Guy Amster, Monty Slatkin, David Reich, Nick Patterson, Christopher Adams, and Iain Mathieson for helpful discussions. We thank Monty Slatkin, Ben Voight, Laurits Skov, and Giulio Genovese for their comments on the manuscript. We thank the editors and reviewers, in particular Laurent Duret, for their valuable comments that led to significant improvement of the manuscript. ZG was supported by a Sloan Research Fellowship and NIH R35GM146810. PM was supported by a Sloan Research Fellowship, the Koret-UC Berkeley-Tel Aviv University Initiative in Computational Biology and Bioinformatics and NIH R35GM142978. YL was supported by the Hellman Fellows Fund to PM. NC was supported by the NSF GRFP fellowship. MP was supported by NIH grant GM122975.

## Additional information

### Competing interests

Molly Przeworski: Reviewing editor, *eLife*. The other authors declare that no competing interests exist.

## Funding

| Funder | Grant reference number | Author |
|---|---|---|
| National Institutes of Health | R35GM146810 | Ziyue Gao |
| Alfred P. Sloan Foundation | FG-2021-15702 | Ziyue Gao |
| National Institutes of Health | R35GM142978 | Priya Moorjani |
| Hellman Family Foundation | | Priya Moorjani Yulin Zhang |
| National Institutes of Health | GM122975 | Molly Przeworski |
| National Science Foundation | DGE 2146752 | Nathan Cramer |
| Alfred P. Sloan Foundation | FG-2019-11943 | Priya Moorjani |

The funders had no role in study design, data collection and interpretation, or the decision to submit the work for publication.

## Author contributions

Ziyue Gao, Priya Moorjani, Conceptualization, Resources, Data curation, Formal analysis, Supervision, Funding acquisition, Validation, Visualization, Methodology, Writing – original draft, Project administration, Writing – review and editing; Yulin Zhang, Resources, Data curation, Formal analysis, Writing – review and editing; Nathan Cramer, Formal analysis, Writing – review and editing; Molly Przeworski, Conceptualization, Funding acquisition, Methodology, Writing – review and editing

## Author ORCIDs

Ziyue Gao http://orcid.org/0000-0001-9244-0238
Yulin Zhang http://orcid.org/0009-0002-8899-5411
Molly Przeworski http://orcid.org/0000-0002-5369-9009
Priya Moorjani http://orcid.org/0000-0002-0947-5673

## Decision letter and Author response

Decision letter https://doi.org/10.7554/eLife.81188.sa1
Author response https://doi.org/10.7554/eLife.81188.sa2

# Additional files

## Supplementary files

• MDAR checklist

## Data availability

All data generated or analyzed during this study were based on publicly available datasets like the 1000 Genomes Project. Source data for Figures 1-4 contain the numerical data used to generate the figures. Outputs for Relate analysis performed in this study are available at the following URL: https://doi.org/10.6078/D19B0H.

The following dataset was generated:

| Author(s) | Year | Dataset title | Dataset URL | Database and Identifier |
|---|---|---|---|---|
| Gao Z, Zhang Y, Cramer N, Przeworski M, Moorjani P | 2023 | Data from: Limited role of generation time changes in driving the evolution of mutation spectrum in humans | https://doi.org/10.6078/D19B0H | Dryad, 10.6078/D19B0H |

The following previously published dataset was used:

| Author(s) | Year | Dataset title | Dataset URL | Database and Identifier |
|---|---|---|---|---|
| Byrska-Bishop M, Evani US, Zhao X | 2021 | High coverage 1000 Genomes Project | https://www.ebi.ac.uk/ena/browser/view/PRJEB31736 | EBI, PRJEB31736 |

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
