## [Editor Report]

This important study investigates temporal variation in patterns of germline mutation during the evolution of human populations. Using a compelling approach that controls for the effects of selection and biased gene conversion the authors show that changes in generation time alone cannot explain the joint patterns observed for different mutation types, suggesting that other factors such as genetic modifiers or environmental exposures must have played a role as well. This work will be of broad interest to population geneticists and evolutionary biologists.

---

## [Decision Letter]

**Decision letter after peer review:**

Thank you for submitting your article "Timing and causes of the evolution of the germline mutation spectrum in humans" for consideration by *eLife*. Your article has been reviewed by 2 peer reviewers, and the evaluation has been overseen by a Reviewing Editor and George Perry as the Senior Editor. The following individual involved in review of your submission has agreed to reveal their identity: Laurent Duret (Reviewer #1).

Essential revisions:

Both reviewers and the editor agree that this is an interesting piece of work. However, several additional controls are required to strengthen the robustness of the results (in particular regarding the possible impact of polarization errors – notably at CpGs, and regarding the reliability of Relate dating estimates). The specific analyses and revisions we would like to see are laid out in detail in the two reviews.

In addition, we had some concerns about the plausibility of the archaic introgression hypothesis for explaining why the ratio of T>C over T>G differs significantly in African samples compared to non-African samples among mutations that are estimated to be much older than the out-of-Africa migration. We wonder whether it would be possible to actually estimate what amount of introgression was needed to account for this signal. We realize that this might be tricky to answer since we don't know the precise mutational signature of the archaic species. But maybe the authors have some ideas. Alternatively, the authors should provide a more detailed discussion of the introgression hypothesis to make it more clear (and explain why ND10 and ND01 variants could behave differently from ND11 variants).

*Reviewer #1 (Recommendations for the authors):*

This manuscript reports very interesting observations, but several additional tests have to be done to check whether the African-specific variation in the ratio of T>C/T>G observed among old variants is real or if it might result from methodological artefacts.

Notably, it is not clear to me if the reported pattern is driven by variants that are specific to the African samples, or if it also observed among variants that are shared across populations (which would point to a problem in the dating of mutations). Furthermore, I suspect that polarization errors (notably at CpG sites) might be responsible for the reported pattern. My comments are developed below.

It is possible that I misunderstood something, but in any case, I think these points need to be clarified before this manuscript can be published. There are also several important points of the methodology that need to be explained in more detail. Finally, I included several other suggestions that might be helpful to improve the manuscript.

1. Shared vs private variants

As I understand, the authors first applied Relate to the entire the 1000 Genomes project dataset (N~2,500 individuals?? This should be indicated in the methods). Then, for each population, they inferred the mutation ages by splitting the Relate output genealogies into subtrees for each population and re-estimated the branch lengths to obtain the final mutation ages. Many variants are shared across populations. Thus, if I understood correctly, each shared variant has multiple age estimates (one per population in which the derived allele has been sampled), that can be considered as replicate estimates (in an ideal world, they should be the same, but they may differ because of limited signal in the data and of simplifying assumptions in the methods). What is not clear to me is how these shared variants were considered by the authors. Did they focus their analyses on variants that are private to each pop? Or did they include all shared variants in their analyses?

Given that they do not mention this point, I presume that they took this latter option. I guess that a large fraction of the old variants are shared across populations. For the sake of my demonstration, let us imagine an extreme case, where old variants (say >20,000 generations) would all be shared across human pops. Imagine that the T>C over T>G ratio changed abruptly 50,000 generations ago. In principle, the exact same shift should be detected at this epoch, whatever the present-day population analyzed. However, there is a large uncertainty around age estimates (in particular for old mutations), and if this uncertainty is larger in some populations than in others (e.g. due to pop demographic history), then the signal for this mutational change might differ, both in intensity and in estimated timing, across populations. Thus, this could potentially contribute to the reported differences in 'old' mutational patterns across populations.

To test this, the authors should repeat their analysis of the T>C over T>G ratio (Figure 2A), specifically on variants that are shared across the 3 pops considered [NB: the analysis of non-ND11 sites in Figure 2D suggests that their results are robust to this potential bias; however, I would prefer to see a direct test]. Conversely, the signal for an old shift in mutation pattern should be much stronger if they focus their analyses on variants that are private to each population. I think it would be really useful to present these two analyses so that we can understand the source of the pattern.

Moreover, to be able to evaluate the impact of the different variant filtering criteria that they use, I think it would be helpful to provide information on the number of SNPs analyzed in each panel of Figure 2 (and associated SupFigure 2.xx), and also on the proportion of shared/private SNPs per age bin.

2. Mutation polarization errors

To verify that the peculiar patterns of pop-specific T>C/T>G mutation ratio they observe in old alleles do not stem from inaccuracies in the polarization of ancestral and derived alleles, the authors repeated the analysis by determining the ancestral state on the basis of the chimpanzee reference genome.

Although the results are qualitatively similar (SupFigure 2.7A), I was surprised to see that they are quantitatively quite different: the difference in T>C/T>G ratio between African and non-African samples is much stronger when variants are polarized with the chimpanzee reference genome than when they are polarized with the '6-EPO human ancestral genome' (Figure 2A). Is it simply due to the fact that more SNPs can be polarized with the chimpanzee than with the EPO ancestral genome? (it would be useful to report sample sizes in these figures). If not, then this would imply that polarization errors do have an important impact on the observed pattern (which would weaken the conclusions of the authors).

To check that, the authors should repeat the analyses with a common set of SNPs (for which the ancestral state was inferred by both methods), and test to what extent the patterns differ according to the polarization method.

The authors should also provide information on how the '6-EPO human ancestral genome' was inferred (indicate the species included in the EPO alignment, and the principle of the method that was used to infer ancestral states), and how they used it. Notably, the EPO ancestral genome makes the distinction between sites for which the ancestral state is considered of 'high-confidence' (reported in upper case) or 'low-confidence' (lower case). Did the author use all sites, or only the high-confidence ones (this should be indicated in the Methods)? Does this make a difference if the analysis is restricted to low-confidence or high-confidence sites?

3. Mutations in CpG or non-CpG context

CpG sites are mutational hotspots and are therefore particularly prone to recurrent mutations and hence to polarization errors. The way the authors handled CpG mutation is not very well detailed. They wrote (line 141): 'we divided C>T SNPs into sub-types occurring in CpG and non-CpG contexts by considering the flanking base pair on either side of the variant'.

If I understand correctly, for the reverse mutation type (T>C), they did not distinguish CpG vs. non-CpG contexts. This implies that mis-polarized C>T CpG mutations are included in the dataset of T>C variants. I imagine that polarization errors may have an important impact on the inference of mutation age. If mis-polarized recent C>T CpG mutations tend to be inferred as old T>C mutations, then the difference in T>C/T>G old mutations observed between African and non-African might simply be due to differences in the total number of variants (and hence in the number of mis-polarized variants) across pops. NB: this would explain why the signal disappears when ND11 sites (which potentially correspond to polarization errors) are excluded – but not ND10 or ND01 (which are more likely to correspond to bona fide old mutations).

To check that, the authors should repeat their analyses of T>C/T>G ratio, after having excluded all variants that potentially arose in a CpG context (i.e. for which either the REF or the ALT allele is in a CpG context – whatever the inferred ancestral state).

4. gBGC

The authors carefully accounted for the possible confounding effects of gBGC. However, the way they explain this point in the results (lines 187 to 202) is unclear. They wrote (line 194 p. 5): "Assuming no systematic evolution in the relative rates of S>W mutations and W>S mutations and unbiased estimation of allele age under gBGC, we would expect similar fractions of S>W and W>S mutations across age bins". This statement is incorrect: if the genome is subject to gBGC, then the ratio W>S/S>W is expected to increase with alleles age; and if the genome is not subject to gBGC, then it does not make sense to refer to an 'unbiased estimation of allele age under gBGC'.

I would recommend the authors to reorganize this section :

1. State that because of gBGC, the relative proportion of W>S vs S>W variants is expected to increase with the age of alleles.

2. Accordingly, they observe an enrichment of W>S variants relative to S>W variants in older age bins (SupFigure 1.6). According to the gBGC model, this enrichment should be positively correlated with recombination rate. The authors have not performed this latter analysis, but I suggest they should; this would be a useful positive control of the signature of gBGC, that they can contrast with what they see in SupFigure 2.3 (where they checked that the patterns they observed are robust to variation in recombination rate, so that to exclude any bias that could be caused by gBGC).

3. Conclude that it is important to take gBGC into account.

To avoid biases that might be caused by gBGC, the authors computed ratios of mutation rates, for pairs of mutation types that are a priori expected to be either not affected by gBGC (C>G/T>A) or equally affected by gBGC (e.g. T>C/T>G). It should be noted however that the mismatch repair mechanisms underlying gBGC (which are not known yet) might act differently on different types of mismatches. For instance, in bacteria, the MutY Adenine DNA-glycosylase of the BER is more efficient on transversion than on transition mismatches (Tsai-Wu et al., 1992 doi: 10.1073/pnas.89.18.8779). In humans, gBGC is stronger on W:S heterozygous sites that are in a CpG context compared to non-CpGs (Halldorsson et al., 2019). It is therefore in principle possible that gBGC contributes to variation in the ratios T>C/T>G or C>T/C>A across age bins. The authors mention this point in their discussion (line 675), but I think it would be useful to mention it also in the result section.

To assess such possible effects, the authors checked that the patterns they observed are robust to variation in recombination rate (SupFigure 2.3). This control is essential, and this is the reason why I think it would be important to demonstrate the efficiency of this test by showing the signature of gBGC on W>S vs S>W variants (see above).

*Reviewer #2 (Recommendations for the authors):*

1. Line 595 says that the TCC signal "may not be specific to Europeans." It's probably worth strengthening this to note that we know it's also in South Asians and has been localized to the descendants of an ancient Anatolian population.

2. It's very interesting that the East Asian T>A enrichment appears as a robust signal given that the prior association of the Japanese T>A enrichment with the Anderson-Trocmé, et al. cell line artifact. Can the authors comment on whether they find evidence of an authentic enrichment in any of the sequence contexts previously associated with the cell line artifact?

3. In line 646, the authors mention that the deeply diverged Khoe-San lineage is a place where we might find ancient mutation spectrum differences. I believe Do, et al. Nature Genetics 2015 in fact did find a mutation spectrum difference between the San and other Africans.

4. I like the "implications" paragraph (lines 690-701) and agree that much more needs to be done to jointly model mutation spectrum evolution with demographic history. It'd be good to note that some work along these lines has already been reported in the original Relate paper as well as DeWitt, et al. 2021 and Speidel, et al. 2021, including evidence of the type of confounding the authors discuss.

---

## [Author Response]

Reviewer #1 (Recommendations for the authors):This manuscript reports very interesting observations, but several additional tests have to be done to check whether the African-specific variation in the ratio of T>C/T>G observed among old variants is real or if it might result from methodological artefacts.Notably, it is not clear to me if the reported pattern is driven by variants that are specific to the African samples, or if it also observed among variants that are shared across populations (which would point to a problem in the dating of mutations). Furthermore, I suspect that polarization errors (notably at CpG sites) might be responsible for the reported pattern. My comments are developed below.It is possible that I misunderstood something, but in any case, I think these points need to be clarified before this manuscript can be published. There are also several important points of the methodology that need to be explained in more detail. Finally, I included several other suggestions that might be helpful to improve the manuscript.1. Shared vs private variantsAs I understand, the authors first applied Relate to the entire the 1000 Genomes project dataset (N~2,500 individuals?? This should be indicated in the methods). Then, for each population, they inferred the mutation ages by splitting the Relate output genealogies into subtrees for each population and re-estimated the branch lengths to obtain the final mutation ages. Many variants are shared across populations. Thus, if I understood correctly, each shared variant has multiple age estimates (one per population in which the derived allele has been sampled), that can be considered as replicate estimates (in an ideal world, they should be the same, but they may differ because of limited signal in the data and of simplifying assumptions in the methods). What is not clear to me is how these shared variants were considered by the authors. Did they focus their analyses on variants that are private to each pop? Or did they include all shared variants in their analyses?

As the reviewer notes, Relate estimates two mutation ages, one initial estimate based on the entire dataset and a more refined estimate for each population. For most analyses in the manuscript, we used the refined estimates (recommended by Relate) based on all variants observed in each population, including the shared variants and those that are private to some groups. For shared variants, we have now included Figure 2 —figure supplement 7 that compares mutation ages inferred using the entire dataset and those based on the population-specific runs. We found systematic differences in the initial and refined mutation ages, though the population-specific estimates for the same variant found in two populations appeared to have overlapping age ranges for over 90% of the variants.

Given that they do not mention this point, I presume that they took this latter option. I guess that a large fraction of the old variants are shared across populations. For the sake of my demonstration, let us imagine an extreme case, where old variants (say >20,000 generations) would all be shared across human pops. Imagine that the T>C over T>G ratio changed abruptly 50,000 generations ago. In principle, the exact same shift should be detected at this epoch, whatever the present-day population analyzed. However, there is a large uncertainty around age estimates (in particular for old mutations), and if this uncertainty is larger in some populations than in others (e.g. due to pop demographic history), then the signal for this mutational change might differ, both in intensity and in estimated timing, across populations. Thus, this could potentially contribute to the reported differences in 'old' mutational patterns across populations.To test this, the authors should repeat their analysis of the T>C over T>G ratio (Figure 2A), specifically on variants that are shared across the 3 pops considered [NB: the analysis of non-ND11 sites in Figure 2D suggests that their results are robust to this potential bias; however, I would prefer to see a direct test]. Conversely, the signal for an old shift in mutation pattern should be much stronger if they focus their analyses on variants that are private to each population. I think it would be really useful to present these two analyses so that we can understand the source of the pattern.

Indeed, a substantial fraction of old variants inferred to predate out-of-Africa migration are broadly shared across populations (defined as segregating across continental groups and present in at least one population from each continental group), but a large fraction (~45%– 65% depending on the population) of old variants are non-shared (i.e., present in one or two continental groups), possibly because the polymorphism was lost in other groups or is present at low frequency and thus undetected in 1000G samples.

Following the reviewer’s suggestion, we repeated analysis of the T>C/T>G ratio in shared and non-shared variants separately (Figure 2—figure supplement 8). For shared variants, we observed no significant inter-population differences in the T>C/T>G ratio, despite some elevation in old variants compared to younger variants in all populations; in contrast, the inter-population differences for non-shared variants are highly significant, suggesting that the T>C / T>G signal is driven mostly by non-shared variants instead of bias or inaccuracy in allele age estimation.

Moreover, to be able to evaluate the impact of the different variant filtering criteria that they use, I think it would be helpful to provide information on the number of SNPs analyzed in each panel of Figure 2 (and associated SupFigure 2.xx), and also on the proportion of shared/private SNPs per age bin.

We agree with the reviewer that the numbers of SNPs in each panel are useful for readers to get a sense of relative power in different analyses. However, including the SNP count of each mutation type, in each population, in each age bin within the figure, makes it very crowded and hard to follow. Therefore, we have included separate tables with this information. We had previously provided the pseudo-counts of 96 trinucleotide mutation types in each time window in Figure 1 Source Data 2 that can be used to compute the total number of SNPs in Figure 2 A, B and D. We have now added Figure 2 Source Data, which contains the pseudocounts of mutations classified into eight types in the commonly accessible region–– including, excluding, and within the maternal mutation hotspots. For a subset of figure supplements (e.g., those describing the comparisons between shared and non-shared variants), we have also added the number of SNPs across certain age bins in the figure for easy reference.

2. Mutation polarization errorsTo verify that the peculiar patterns of pop-specific T>C/T>G mutation ratio they observe in old alleles do not stem from inaccuracies in the polarization of ancestral and derived alleles, the authors repeated the analysis by determining the ancestral state on the basis of the chimpanzee reference genome.Although the results are qualitatively similar (SupFigure 2.7A), I was surprised to see that they are quantitatively quite different: the difference in T>C/T>G ratio between African and non-African samples is much stronger when variants are polarized with the chimpanzee reference genome than when they are polarized with the '6-EPO human ancestral genome' (Figure 2A). Is it simply due to the fact that more SNPs can be polarized with the chimpanzee than with the EPO ancestral genome? (it would be useful to report sample sizes in these figures). If not, then this would imply that polarization errors do have an important impact on the observed pattern (which would weaken the conclusions of the authors).To check that, the authors should repeat the analyses with a common set of SNPs (for which the ancestral state was inferred by both methods), and test to what extent the patterns differ according to the polarization method.

We thank the reviewer for noticing this issue. The difference is probably explained by a choice that we made that was not carefully described in the Methods. Because we only had access to the chimpanzee genome in hg19 coordinates, for this analysis alone, we use the 1000G Phase 3 data which is in hg19. We had separately confirmed that the results for 1000G Phase 3 and 1000G high coverage were qualitatively similar (results not included in the manuscript).

Given the reviewers confusion, we have now performed all the analysis with 1000G high coverage data. We lifted over 1000G high coverage data to hg19 and used the chimpanzee reference genome to assign the ancestral allele (Note, we cannot use the output of Relate directly since it is useful to run the entire analysis with the ancestral genome for reliable comparison). There are indeed ~10-18% more SNPs that can be polarized with the chimpanzee reference genome than with the EPO ancestral genome, which partially explains the stronger inter-population differences based on the former. We have updated the Figure 2 —figure supplement 12A with the new results.

The authors should also provide information on how the '6-EPO human ancestral genome' was inferred (indicate the species included in the EPO alignment, and the principle of the method that was used to infer ancestral states), and how they used it. Notably, the EPO ancestral genome makes the distinction between sites for which the ancestral state is considered of 'high-confidence' (reported in upper case) or 'low-confidence' (lower case). Did the author use all sites, or only the high-confidence ones (this should be indicated in the Methods)? Does this make a difference if the analysis is restricted to low-confidence or high-confidence sites?

For our analysis, we used both the high and low confidence EPO sites. We have added an analysis (Figure 2—figure supplement 12B) including only SNPs that can be polarized with “high-confidence” ancestral alleles. Since there are much fewer low-confidence sites, the inference based on low-confidence sites only is very noisy, so we do not include these results.

3. Mutations in CpG or non-CpG contextCpG sites are mutational hotspots and are therefore particularly prone to recurrent mutations and hence to polarization errors. The way the authors handled CpG mutation is not very well detailed. They wrote (line 141): 'we divided C>T SNPs into sub-types occurring in CpG and non-CpG contexts by considering the flanking base pair on either side of the variant'.If I understand correctly, for the reverse mutation type (T>C), they did not distinguish CpG vs. non-CpG contexts. This implies that mis-polarized C>T CpG mutations are included in the dataset of T>C variants. I imagine that polarization errors may have an important impact on the inference of mutation age. If mis-polarized recent C>T CpG mutations tend to be inferred as old T>C mutations, then the difference in T>C/T>G old mutations observed between African and non-African might simply be due to differences in the total number of variants (and hence in the number of mis-polarized variants) across pops. NB: this would explain why the signal disappears when ND11 sites (which potentially correspond to polarization errors) are excluded – but not ND10 or ND01 (which are more likely to correspond to bona fide old mutations).To check that, the authors should repeat their analyses of T>C/T>G ratio, after having excluded all variants that potentially arose in a CpG context (i.e. for which either the REF or the ALT allele is in a CpG context – whatever the inferred ancestral state).

We are grateful to the reviewer for this comment. While the manuscript was under review, we did an analysis to identify the sequence context in which the T>C signal is enriched by applying non-negative matrix factorization (NMF). We observed that the signal is enriched in TpG context (Figure 2—figure supplement 10). In addition, the three oldest mutation age bins (ages > 28,000 generations) contain substantially greater fractions of TpG>CpG mutations among T>C SNPs compared to the younger bins (Figure 2—figure supplement 11). Notably, the removal of TpGs sites from the analysis shifts the mutation ratio (throughout all age bins), and the T>C/T>G signal is no longer significant (i.e., there is no increase in the ratio in the older bins or population differences among CEU, CHB and YRI; Figure 2D in revised version).

Given that the T>C/T>G is entirely driven by TpG sites that may be prone to polarization errors due to recurrent mutations, our previous hypothesis for this signal becomes very tentative. In light of the technical issues, we refrain from discussing the biological mechanisms related to this signal and hence remove any discussion about archaic population structure or evolution of T>C mutation rates across populations. Instead, we focus on describing the technical issues we uncovered and present this analysis as a cautionary tale for readers and other scientists interested in comparisons of mutation spectrum in humans and other species.

4. gBGCThe authors carefully accounted for the possible confounding effects of gBGC. However, the way they explain this point in the results (lines 187 to 202) is unclear. They wrote (line 194 p. 5): "Assuming no systematic evolution in the relative rates of S>W mutations and W>S mutations and unbiased estimation of allele age under gBGC, we would expect similar fractions of S>W and W>S mutations across age bins". This statement is incorrect: if the genome is subject to gBGC, then the ratio W>S/S>W is expected to increase with alleles age; and if the genome is not subject to gBGC, then it does not make sense to refer to an 'unbiased estimation of allele age under gBGC'.I would recommend the authors to reorganize this section:1. State that because of gBGC, the relative proportion of W>S vs S>W variants is expected to increase with the age of alleles.2. Accordingly, they observe an enrichment of W>S variants relative to S>W variants in older age bins (SupFigure 1.6). According to the gBGC model, this enrichment should be positively correlated with recombination rate. The authors have not performed this latter analysis, but I suggest they should; this would be a useful positive control of the signature of gBGC, that they can contrast with what they see in SupFigure 2.3 (where they checked that the patterns they observed are robust to variation in recombination rate, so that to exclude any bias that could be caused by gBGC).3. Conclude that it is important to take gBGC into account.

We thank the reviewer for the suggestion and have reorganized this section accordingly. We stratified the genome into three bins based on the HapMap recombination map and quantified the enrichment of W>S mutations as a function of allele age (Figure 1—figure supplement 7). Qualitatively, we do observe more variation in the fractions of W>S and S>W mutations across age bins and stronger enrichment of W>S relative S>W in older variants in genomic regions with high recombination rates, but the same pattern is also visible in regions with low recombination rates. The lack of strong contrast between high- and lower combination regions might be partly due to the rapid evolution of recombination hotspots (Coop and Przeworski 2007). Regardless, the differential enrichment of W>S and S>W variants with allele age highlights the need to account for the effect of gBGC in comparing mutation spectra.

To avoid biases that might be caused by gBGC, the authors computed ratios of mutation rates, for pairs of mutation types that are a priori expected to be either not affected by gBGC (C>G/T>A) or equally affected by gBGC (e.g. T>C/T>G). It should be noted however that the mismatch repair mechanisms underlying gBGC (which are not known yet) might act differently on different types of mismatches. For instance, in bacteria, the MutY Adenine DNA-glycosylase of the BER is more efficient on transversion than on transition mismatches (Tsai-Wu et al., 1992 doi: 10.1073/pnas.89.18.8779). In humans, gBGC is stronger on W:S heterozygous sites that are in a CpG context compared to non-CpGs (Halldorsson et al., 2019). It is therefore in principle possible that gBGC contributes to variation in the ratios T>C/T>G or C>T/C>A across age bins. The authors mention this point in their discussion (line 675), but I think it would be useful to mention it also in the result section.

We agree with the reviewer that differential strength of gBGC across mismatch types is in principle possible, but the limited data on non-crossover gene conversion events in mice suggest roughly similar conversion rates across different types of mismatches with C as the recipient and across those with T as the recipient (Figure 4C in Li et al., 2019). We have now included a brief discussion of this possibility in the Results section.

To assess such possible effects, the authors checked that the patterns they observed are robust to variation in recombination rate (SupFigure 2.3). This control is essential, and this is the reason why I think it would be important to demonstrate the efficiency of this test by showing the signature of gBGC on W>S vs S>W variants (see above).

We thank the reviewer for the comment, and we hope the new Figure 1—figure supplement 7 (fractions of S>W, W>S, W>W and W>W variants in low and high recombination rate regions) addresses his concerns.

Reviewer #2 (Recommendations for the authors):1. Line 595 says that the TCC signal "may not be specific to Europeans." It's probably worth strengthening this to note that we know it's also in South Asians and has been localized to the descendants of an ancient Anatolian population.

We agree with the reviewer and have fixed this point in the introduction.

2. It's very interesting that the East Asian T>A enrichment appears as a robust signal given that the prior association of the Japanese T>A enrichment with the Anderson-Trocmé, et al. cell line artifact. Can the authors comment on whether they find evidence of an authentic enrichment in any of the sequence contexts previously associated with the cell line artifact?

We believe the T>A enrichment that we detected in East Asians is unrelated to the cell line artifacts reported by Anderson-Trocmé et al. for several reasons. First, throughout the manuscript, we used high-coverage 1000 Genomes data, which greatly reduces the possibility of calling cell line artifacts. Second, the lower C>G/T>A ratio that we observe is not Japanese-specific and instead shared by at least two East Asian populations (JPT and CHB); additionally, the enrichment is not driven by rare variants. Lastly, the cell line artifacts that Anderson-Trocmé et al. found to be associated with *Q* value were primarily enriched in *AC >*CC sequence context, which is unrelated to the mutation types involved in our T>A signal.

3. In line 646, the authors mention that the deeply diverged Khoe-San lineage is a place where we might find ancient mutation spectrum differences. I believe Do, et al. Nature Genetics 2015 in fact did find a mutation spectrum difference between the San and other Africans.

We were not able to find any comparisons of the mutation spectrum in Do et al. Nature Genetics. In addition, in light of the new results regarding TpG sites, we have removed the discussion of the ancient signal of T>C/T>G ratio entirely so the discussion of mutation spectrum difference between San and other Africans is no longer relevant.

4. I like the "implications" paragraph (lines 690-701) and agree that much more needs to be done to jointly model mutation spectrum evolution with demographic history. It'd be good to note that some work along these lines has already been reported in the original Relate paper as well as DeWitt, et al. 2021 and Speidel, et al. 2021, including evidence of the type of confounding the authors discuss.

We have now revised this paragraph to better reflect the conclusions and challenges in our analysis (see below), which no longer focuses on joint modeling of demographic and mutation spectrum evolution:

“The mutation spectrum of polymorphisms is a convolution of multiple evolutionary forces: mutation, recombination (including gene conversion), natural selection, and their interplay with demography. In this study, we investigated the contribution of these forces to differences in the mutation spectrum across contemporary human populations. For future studies aiming to understand the evolution of mutagenesis based on analyses of polymorphism patterns, it will be crucial to consider more realistic mutation models (including using context-dependent models) for ancestral allele reconstruction and accounting for the impact of non-mutational evolutionary forces. ”

However, we highlight the contributions of other studies in Introduction and Discussion sections.